# STaR: Self-Taught Reasoner
## Bootstrapping Reasoning With Reasoning

**Eric Zelikman**[*1], **Yuhuai Wu**[*12], **Jesse Mu**[1], **Noah D. Goodman**[1]
[1]Department of Computer Science, Stanford University
[2] Google Research
{ezelikman, yuhuai, muj, ngoodman}@stanford.edu

## Abstract

Generating step-by-step "chain-of-thought" rationales improves language model performance on complex reasoning tasks like mathematics or commonsense question-answering. However, inducing language model rationale generation currently requires either constructing massive rationale datasets or sacrificing accuracy by using only few-shot inference. We propose a technique to iteratively leverage a small number of rationale examples and a large dataset without rationales, to bootstrap the ability to perform successively more complex reasoning. This technique, the "Self-Taught Reasoner" (STaR), relies on a simple loop: generate rationales to answer many questions, prompted with a few rationale examples; if the generated answers are wrong, try again to generate a rationale given the correct answer; fine-tune on all the rationales that ultimately yielded correct answers; repeat. We show that STaR significantly improves performance on multiple datasets compared to a model fine-tuned to directly predict final answers, and performs comparably to fine-tuning a $30\times$ larger state-of-the-art language model on CommensenseQA. Thus, STaR lets a model improve itself by learning from its own generated reasoning.[1]

## 1 Introduction

Human decision-making is often the result of extended chains of thought [1, 2]. Recent work has shown that explicit intermediate reasoning ("rationales") can improve large language model (LLM) performance as well [3–8]. For example, [5] demonstrated that LLMs explicitly trained to use "scratchpads" for intermediate steps can attain perfect in-distribution performance on arithmetic, and strong out-of-distribution generalization, while models trained to predict answers directly fail to do either. These works suggest that generating explicit rationales before giving a final answer ("rationale generation") is valuable for LLMs across diverse tasks including mathematical reasoning, commonsense reasoning, code evaluation, social bias inference, and natural language inference. However, the two primary methods for inducing rationale generation both have serious drawbacks.

One approach to rationale generation is the construction of a fine-tuning dataset of rationales, either manually by human annotators or automatically with hand-crafted templates [3–5, 9]. Manual methods are expensive, and it is infeasible to construct such a dataset for each interesting problem [3]. Meanwhile, template-based methods rely on automatically-generated rationales but only work when a general solution is already known [5] or reasonable hard-coded heuristics can be made [4].

An alternative is to leverage in-context learning by including only a few rationale examples in the language model prompt. This has been shown to improve accuracy on mathematical and symbolic reasoning tasks relative to prompting without rationales ("direct" prompting) [5, 6]. Yet, while few-shot techniques with rationales tend to outperform their non-reasoning counterparts, they generally substantially underperform models fine-tuned to directly predict answers using larger datasets [5, 6].

---

[*]These authors contributed equally to this work
[1]We release our code at https://github.com/ezelikman/STaR.

36th Conference on Neural Information Processing Systems (NeurIPS 2022).

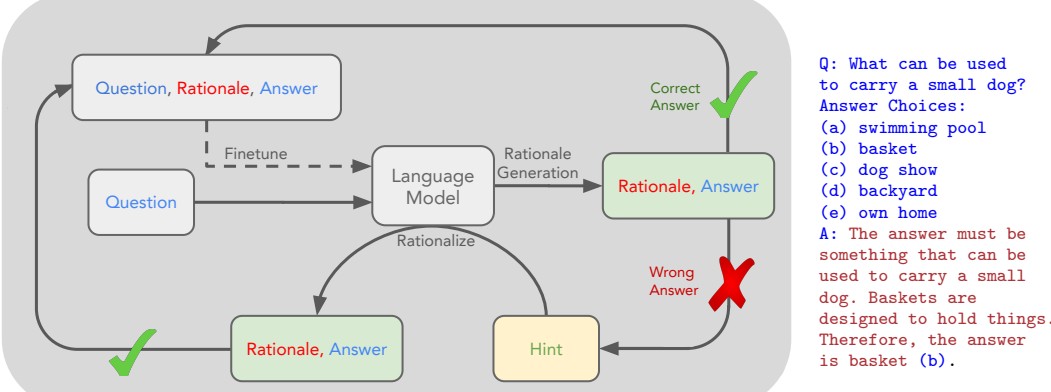

Figure 1: An overview of STaR and a STaR-generated rationale on CommonsenseQA. We indicate the fine-tuning outer loop with a dashed line. The questions and ground truth answers are expected to be present in the dataset, while the rationales are generated using STaR.

In this paper, we adopt a different approach: by leveraging the LLM's pre-existing reasoning ability, we iteratively *bootstrap* the ability to generate high-quality rationales. Specifically, we few-shot prompt a large language model to self-generate rationales and refine the model's ability further by fine-tuning on those rationales that lead to correct answers. We repeat this procedure, using the improved model to generate the next training set each time. This is a synergistic process, where improvements in rationale generation improve the training data, and improvements in training data further improve rationale generation.

However, we find this loop eventually fails to solve any new problems in the training set because it receives no direct training signal for problems it fails to solve. To overcome this issue, we propose **rationalization**: for each problem that the model fails to answer correctly, we generate a new rationale by providing the model with the correct answer. This lets the model reason backward—given the correct answer, the model can more easily generate a useful rationale. These rationales are then collected as part of the training data, which often improves overall accuracy.

We thus develop the Self-Taught Reasoner (STaR, Fig. 1) method, a scalable bootstrapping method allowing models to learn to generate their own rationales, while also learning to solve increasingly difficult problems. In our method, we repeat the following process: in each iteration, first construct a finetuning dataset by attempting to solve the dataset using the current model's **rationale generation** ability; then, augment this dataset using **rationalization**, justifying ground-truth answers to problems the model failed to solve; finally, finetune the large language model on the combined dataset.

Applying STaR on arithmetic, math word problems, and commonsense reasoning, we observe it is able to effectively translate a small number of few-shot prompts into a large rationale dataset, yielding dramatic performance improvements. On CommonsenseQA [10], we find STaR improves over both a few-shot baseline (+35.9%) and a baseline fine-tuned to directly predict answers (+12.5%) , and performs comparably to a fine-tuned model that is 30× larger (72.5% vs. 73.0%).

Thus, we make the following contributions:

1. We propose a bootstrapping mechanism to iteratively generate a rationale dataset from a few initial examples with rationales—without needing to check new rationales' correctness.

2. We complement **rationale generation** with **rationalization**, where a model is tasked with justifying an answer and then fine-tuned as if it had come up with the rationale without any hint. We show rationalization accelerates and improves the bootstrapping process.

3. We evaluate these techniques with a variety of ablations in both mathematical and commonsense reasoning domains.

4. We propose what is, to our knowledge, the first technique to allow a pre-trained large language model to iteratively use its language modeling capacity to improve itself.

## 2 Background and Related Work

**In-context Learning**   Recently, a collection of works has emerged exploring the capacity for large language models to perform in-context learning [11, 12]. In essence, in-context learning treats few-shot learning as a language modeling problem, by showing a few examples in the context (i.e. prompt), and allowing the model to learn and identify the pattern to apply to new examples. Some have studied in-context learning based on the language modeling objective in terms of Bayesian inference [13] while others have attempted to describe the process more mechanistically in terms of "induction heads" [14]. Moreover, differences in prompt configurations have been known to have dramatic effects on few-shot performance. Some have even found that replacing few-shot prompts with a "soft prompt" which can be optimized in embedding space results in noticeable gains [15]. Instead of emphasizing the representation of the question, we focus on the model output; in particular, we focus on the model's ability to reason through a problem before coming to a conclusion.

**Rationales**   One initial work on the impact of rationales on language model performance was [3], showing that training a language model on a dataset with explicit rationales preceding the answer could improve a model's ability to generate the final answer. However, this required many thousands of training examples to be manually annotated with human reasoning. Recently, [5] demonstrated that step-by-step "scratchpads" improve fine-tuned LLM performance and generalization on tasks such as arithmetic, polynomial evaluation, and program evaluation. Similarly, [6] used a single few-shot "chain-of-thought" reasoning prompt to improve model performance on tasks without fine-tuning. Finally, [16] showed that a curriculum learning approach could help solve formal math problems, if 1) they were translated into Lean (a theorem-proving language [17]), 2) one could directly evaluate the proofs' validity, 3) one could sample many solutions per problem, 4) had trained a separate value function model, and 5) started with GPT-f (a model fine-tuned on a large math dataset [18]). We note there are many domains where these conditions do not all apply. In addition, works have aimed to explain why rationales help: some have analyzed their impact from the perspective of latent variable models [19] while others have provided formal proofs of the benefit of intermediate task supervision [20].

**Iterated Learning**   A variety of iterated learning algorithms have been proposed, where solutions or successful methods which are found are in turn used to find additional solutions [21, 22, 16]. [21] introduced Expert Iteration (ExIt), a reinforcement learning technique serving as an inspiration for our approach. Essentially, it consists of a loop of self-play by an "apprentice," followed by imitation learning with feedback from a slower "expert" and then the replacement of the expert with the now-improved apprentice. [16] builds off of ExIt for formal reasoning, while [22] applies iterated learning to visual question answering using modular networks which can be combined compositionally. There are further similarities between STaR and expert iteration methods [21]. For example, filtering generated examples based on whether their ultimate answer matches the target can be seen as expert feedback. However, we have a fixed "expert" and do not train a separate value function. The idea of alternating between filtering steps and training steps more broadly is also well-grounded in prior work in NLP such as [23] and in other weak supervision contexts [24, 25].

**Natural Language Explanations**   Natural language explanations have also been discussed from the perspective of explainable machine learning, focusing on justification rather than reasoning [26, 27]. The motivation for this line of work is largely grounded in explainable decision making, and similarly to [3], generally does not find that requiring post-hoc explanations improves model performance.

## 3 Method

### 3.1 Rationale Generation Bootstrapping (STaR Without Rationalization)

We are given a pretrained LLM $M$ and an initial dataset of problems $x$ (including answer choices if applicable) with correct final answers $y$: $\mathcal{D} = \{(x_i, y_i)\}_{i=1}^{D}$. Our technique starts with a small *prompt* set $\mathcal{P}$ of examples with intermediate *rationales* $r$: $\mathcal{P} = \{(x_i^p, r_i^p, y_i^p)\}_{i=1}^{P}$, where $P \ll D$ (e.g. $P = 10$). Like standard few-shot prompting, we concatenate this prompt set to each example in $\mathcal{D}$, i.e. $x_i = (x_1^p, r_1^p, y_1^p, \ldots, x_P^p, r_P^p, y_P^p, x_i)$, which encourages the model to produce a rationale $\hat{r}_i$ for $x_i$ followed by an answer $\hat{y}_i$. We assume that rationales that lead to correct answers are of better quality than those that lead to incorrect answers. Therefore, we filter the generated rationales to include only the ones which result in the correct answer ($\hat{y}_i = y_i$). We fine-tune the base model $M$ on this filtered dataset, and then restart this process by generating the new rationales with the newly fine-tuned model. We keep repeating this process until the performance plateaus. Note that during this process, once we collect a new dataset, we train from the original pre-trained model $M$ instead of continually training one model to avoid overfitting. We provide an outline of this algorithm in Algorithm 1.

STaR can be seen as an approximation to an RL-style policy gradient objective. To see this, note that $M$ can be viewed as a discrete latent variable model $p_M(y \mid x) = \sum_r p(r \mid x)p(y \mid x, r)$; in other words, $M$ first samples a latent rationale $r$ before predicting $y$. Now, given the indicator reward function $\mathbb{1}(\hat{y} = y)$, the total expected reward across the dataset is

$$J(M, X, Y) = \sum_i \mathbb{E}_{\hat{r}_i, \hat{y}_i \sim p_M(\cdot \mid x_i)} \mathbb{1}(\hat{y}_i = y_i), \tag{1}$$

$$\nabla J(M, X, Y) = \sum_i \mathbb{E}_{\hat{r}_i, \hat{y}_i \sim p_M(\cdot \mid x_i)} \left[ \mathbb{1}(\hat{y}_i = y_i) \cdot \nabla \log p_M(\hat{y}_i, \hat{r}_i \mid x_i) \right], \tag{2}$$

$$\tag{3}$$

where the gradient is obtained via the standard log-derivative trick for policy gradients. Note that the indicator function discards the gradient for all sampled rationales that do not lead to the correct answer $y_i$: this is the filtering process in STaR (Line 5). Thus, STaR approximates $J$ by (1) greedily decoding samples of $(\hat{r}_i, \hat{y}_i)$ to reduce variance of this estimate (at the cost of potentially biased exploration of rationales), and (2) taking multiple gradient steps on the same batch of data (similar to some policy gradient algorithms [28]). These approximations make STaR a simple and broadly applicable method that can be implemented with standard LLM training machinery; future work should more closely investigate the link between STaR and the RL objective above.

## 3.2 Rationalization

The rationale generation bootstrapping algorithm carries a limitation. As the model is only trained on the examples which it answers correctly, improvement ends when the model fails to solve new problems in the training set. This is fundamentally due to the fact that the algorithm cannot obtain any training signal from failed examples. Inspired by [3], we propose a technique we call "rationalization". Specifically, we provide the answer as a hint to the model and ask it to generate rationales in the same style as in the previous rationale generation step. Given the answer, the model is able to reason backwards, and hence more easily generate a rationale leading to the correct answer. Figure 2 provides an example of this format on CQA, also used in the few-shot rationalization prompt, indicating that "(b) grocery cart" is the correct answer. We apply rationalization to problems which the model failed to solve with rationale genera-

```
Q: Where do you put your grapes just
before checking out?
Answer Choices:
(a) mouth
(b) grocery cart (CORRECT)
(c) super market
(d) fruit basket
(e) fruit market
A: The answer should be the place
where grocery items are placed before
 checking out. Of the above choices,
grocery cart makes the most sense for
 holding grocery items. Therefore,
the answer is grocery cart (b).
```

Figure 2: A few-shot prompt hint we use for rationalization (and not for rationale generation), using the rationale from [6], with its hint included in green, followed by the rationale and the answer generated by the model.

tion. When adding a rationalization-generated rationale to our dataset, we do not include the hint in its corresponding prompt, as if the model had come up with the rationale without the hint. After filtering, we fine-tune on the previously generated dataset combined with the rationalization-generated dataset.

---

**Algorithm 1** STaR

    **Input** $M$: a pretrained LLM; dataset $\mathcal{D} = \{(x_i, y_i)\}_{i=1}^D$ (w/ few-shot prompts)
1:  $M_0 \leftarrow M$ # Copy the original model
2:  **for** $n$ **in** $1...N$ **do** # Outer loop
3:     $(\hat{r}_i, \hat{y}_i) \leftarrow M_{n-1}(x_i) \quad \forall i \in [1, D]$ # Perform rationale generation
4:     $(\hat{r}_i^{\text{rat}}, \hat{y}_i^{\text{rat}}) \leftarrow M_{n-1}(\text{add\_hint}(x_i, y_i)) \quad \forall i \in [1, D]$ # Perform rationalization
5:     $\mathcal{D}_n \leftarrow \{(x_i, \hat{r}_i, y_i) \mid i \in [1, D] \wedge \hat{y}_i = y_i\}$ # Filter rationales using ground truth answers
6:     $\mathcal{D}_n^{\text{rat}} \leftarrow \{(x_i, \hat{r}_i^{\text{rat}}, y_i) \mid i \in [1, D] \wedge \hat{y}_i \neq y_i \wedge \hat{y}_i^{\text{rat}} = y_i\}$ # Filter rationalized rationales
7:     $M_n \leftarrow \text{train}(M, \mathcal{D}_n \cup \mathcal{D}_n^{\text{rat}})$ # Finetune the original model on correct solutions - inner loop
8:  **end for**

---

Algorithm 1 describes the full algorithm, with the parts in blue corresponding to rationalization. Without those parts, Algorithm 1 corresponds to STaR without rationalization. Figure 1 provides an overview diagram. Fine-tuning on the dataset generated by rationalization has a crucial benefit of exposing the model to difficult problems which otherwise would not have appeared in its finetuning

dataset. This can be understood as challenging the model to "think outside the box" about problems on which it was unsuccessful. A secondary benefit of rationalization is an increase in dataset size.

## 4 Experiments

For our experiments, we focus on arithmetic, commonsense reasoning, and grade school math to demonstrate STaR's breadth. In particular, for arithmetic, we follow a setup inspired by [5]. For commonsense question-answering we follow [13, 6] and use CommonsenseQA (CQA), a widely used multiple-choice dataset for this domain [10]. For grade school math, we use GSM8K from [9].

### 4.1 Experimental Protocol

We used GPT-J as our base language model, and the fine-tuning script from the GPT-J repository [29]. We chose GPT-J, a 6B-parameter model, because the checkpoint and fine-tuning code are publicly available [29], and the model is large enough to generate rationales of non-trivial quality to be bootstrapped from. More hyperparameter details about GPT-J and our fine-tuning are included in Appendix G. Following the default setting of [29], we perform a 100-step learning rate warmup, from which point we use a constant learning rate. Unless stated otherwise, we start with 40 training steps at the first outer loop, and increase the number of inner-loop fine-tuning training steps by 20% with each outer loop. In general, we found that training more slowly at the beginning ultimately benefits model performance. We expect that further improvement is possible via a thorough hyperparameter search—we leave this to future work due to computational constraints.

For arithmetic problems, we first generate a dataset of 50,000 randomly sampled questions (uniformly over the digit lengths) in the format introduced by [5]. For each outer loop iteration on arithmetic, we sample 10,000 problems from the dataset. We use 10 random few-shot rationale examples for each digit for its corresponding few-shot prompt. For each of the $9,741$ questions in the training set of CommonsenseQA, we add the question to the few-shot rationale prompt, and prompt the model to generate the rationale and answer for that question. For few shot prompting on CQA, we start with the same 10 questions as used in [6], with the rationales modified slightly to fix an incorrect answer and to more explicitly reference relevant knowledge. We include these modified prompts in Appendix B[2]. These prompts serve as our complete set of explanations. We run STaR until we see performance saturate, and we report the best results.

When performing rationalization, we find that the choice to include or omit few-shot prompts on outer-loop iterations after the first iteration does not have a substantial impact on the method's ultimate performance. However, there are some nuances which we discuss further in Section 5, leading us to use few-shot prompts unless stated otherwise.

### 4.2 Datasets

**Arithmetic** The arithmetic task is to calculate the sum of two $n$-digit integers. We generate the dataset based on the descriptions in [5] and visualize an example scratchpad in Figure 3. Everything up to and including "`Target:`" is given as part of a prompt, and the model is asked to generate the scratchpad (start/end indicated by "`<scratch>`") and the final answer, as in [5]. Each line of the scratchpad corresponds to the summation of each pair of digits from the final digit to the first digit, the accumulating final digits of the answer, and a carry digit corresponding to whether the previous pair summed to at least 10. We include few-shot prompts for 1 to 5 digits. When performing rationalization, we include the correct answer after "`Target`" and query the model to produce the scratchpad and then reproduce the correct answer following the scratchpad.

```
Input:
6 2 4 + 2 5 9
Target:
<scratch>
6 2 4 + 2 5 9 , C: 0
2 + 5 , 3   C: 1
6 + 2 , 8 3   C: 0
, 8 8 3   C: 0
0 8 8 3
</scratch>
8 8 3
```

Figure 3: A visualization of a 3-digit arithmetic problem with a scratchpad. C corresponds to the carry from the previous digit's summation.

**CommonsenseQA** The multiple-choice commonsense reasoning task, CommonsenseQA [10] (CQA), is constructed from ConceptNet, a semantic graph of concepts and their relationships with over a million nodes [31]. [10] identified a set of "target" concepts in ConceptNet for each question, where the target concepts share a semantic relationship to one "source" concept. Then each question is crowdsourced to allow a reader to identify one target concept, while

---

[2]Based on [30], this is unlikely to meaningfully affect [6]'s few-shot performance.

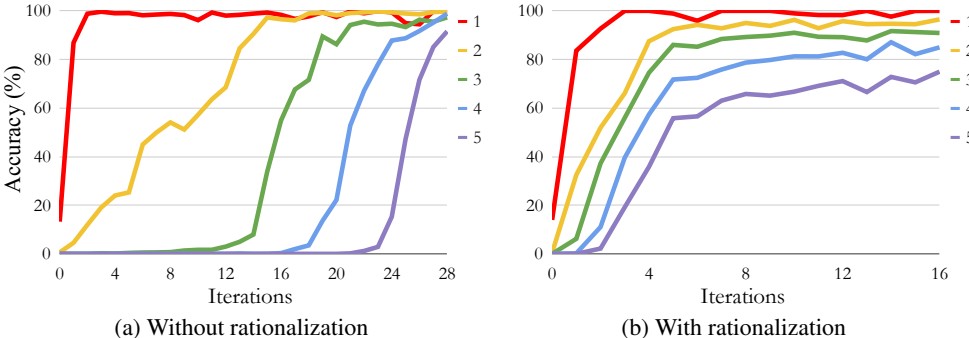

(a) Without rationalization        (b) With rationalization

Figure 4: A visualization of the accuracy of $n$-digit summation with each iteration of STaR with and without rationalization. Each series corresponds to the accuracy of summing two $n$-digit numbers.

mentioning the source concept. In addition, two distractor answers are added. The dataset has 12,247 questions, each with five choices, with 9,741 in the train set, 1,221 in the dev set, and 1,285 in the (withheld) test set.

Corresponding to the broad variety of ConceptNet, CQA contains a diverse set of questions which require commonsense reasoning ability building off of standard world knowledge, where human performance is 89% [10]. Many have pointed out that CQA contains a number of biases, along several dimensions including gender [3]. We discuss how this may impact our method in Section 6. There are also many typos and questions which are fundamentally ambiguous[3]. We use it despite these issues as it is a general question-answering dataset relying on both common world knowledge and simple reasoning, which serves as a good test-bed for our method.

**Grade School Math (GSM8K)**    We also evaluate on the Grade School Math (GSM8K) dataset, containing 7,473 train and 1,319 test examples of grade-school-level word problems [9]. These math problems are posed in natural language and require two to eight calculation steps to arrive at a final answer. This dataset combines the skills needed for arithmetic and commonsense reasoning. For rationalization, we include the final answer in parentheses immediately after the question as a hint.

### 4.3  Symbolic Reasoning: Results on Arithmetic

The accuracies of the model across digits 1-5 over each iteration of the outer loop are plotted in Figure 4. After running STaR for 16 iterations, the overall accuracy is 89.5%. For reference, a baseline trained on 10,000 examples without rationales for 5,000 steps attains 76.3% accuracy. Notably, few-shot accuracy on arithmetic problems is very low, even with rationales: accuracy on 2-digit addition is less than 1%, and accuracy on more digits close to zero.

With rationalization, the accuracy is able to improve especially quickly. After one fine-tuning iteration on the model's generated scratchpads, 2-digit addition improves to 32% from less than 1%. Without rationalization, the performance improvement is stagewise: the model generally has poor performance on the $n$-digit sum until it has good performance on the $(n-1)$-digit sum. With rationalization, the model can learn many lengths at once, though not with equal accuracy. Rationalization allows many problems to be solved few-shot, so we start STaR training with 300 steps (note, doing so without rationalization causes overfitting on 1-digit addition), and increase training by 20 steps per iteration.

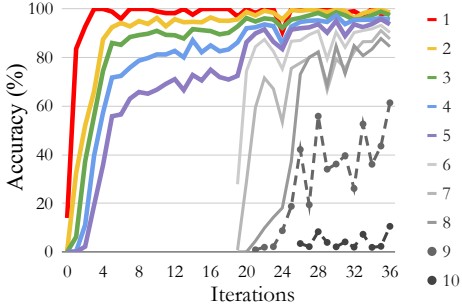

Figure 5: We introduce additional digits to STaR with rationalization at the $20^{th}$ iteration.

We also perform an experiment where we continue pre-training STaR with rationalization with additional digits, starting before the 20th iteration, while keeping the total number of training examples

---
[3]For example, "Billy bought coffee and waited for his wife to arrive from France. Where might he have been?" includes airport and train station as options. The correct answer, perhaps surprisingly, is train station.

Table 1: We evaluate several baselines, including a few-shot GPT-J evaluation both with and without scratchpads, a GPT-J baseline finetuned to directly predict the answer, and STaR with and without rationalization applied to GPT-J. We use CoT to denote non-STaR models outputting rationales, and Direct to indicate those directly predicting the final answer. Note the final STaR model is trained on 78.2% of the training dataset with rationale generation, and an additional 8.5% from rationalization.

| | CQA Dev Set Accuracy (%) | Train Data Used (%) |
|---|---|---|
| *GPT-3 Direct Finetuned [32]* | *73.0* | 100 |
| Few-shot Direct GPT-J | 20.9 | ∼0 |
| Few-shot CoT GPT-J [4] | 36.6 | ∼0 |
| Few-shot CoT LaMDA 137B [6] | 55.6 | ∼0 |
| GPT-J Direct Finetuned | 60.0 | 100 |
| STaR without rationalization | 68.8 | 69.7 |
| STaR with rationalization | **72.5** | 86.7 |

fixed at each iteration. We find that not only does this appear to quickly improve performance on the initial set of digits, but when evaluated on 9 and 10 digit examples, never seen during training, the model successfully solves many of these out-of-distribution problems. As visualized in Figure 5, the introduction of these digits appears to make the training less stable, but the exact cause is unclear.

## 4.4 Natural Language Reasoning: Commonsense Question Answering

The CommonsesenseQA (CQA) setting introduces several new challenges. In the arithmetic task, an incorrect scratchpad in the reasoning step, and to a lesser degree in the rationalization step, was extremely likely to result in an incorrect answer. On the other hand, CQA problems are 5-way multiple choice questions. Thus, one will get the right answer at random approximately 20% of the time, regardless of the quality of reasoning. Moreover, some simple heuristics (e.g. semantic similarity) can meaningfully improve this to ≈30% without any reasoning, as shown by [10].

We evaluate this dataset as described in the experimental protocol and compare to several baselines. The first baseline is to finetune GPT-J to directly output the final answer, which we call "GPT-J Finetuned". We also compare to GPT-3 finetuned to directly predict the final answer from [32], and a 137B parameter Lambda model few-shot prompted with chain-of-thought (CoT) rationales from [6].

We found that, as shown in Table 1, STaR without rationalization outperformed GPT-J fine-tuned directly on the final answer for the entire dataset, despite training on less of the data. The inclusion of rationalization improved this performance to 72.5%, far closer to the 73% of the 30× larger GPT-3. As expected, we also see STaR surpassed the few-shot baselines, including the much-larger 137B LaMDA model [33, 6]. We expect accuracy would be further improved if we applied STaR to a model with higher few-shot performance. Note that substantially higher performance is possible: [32] demonstrated a custom 39-model ensemble which reached "super-human" performance on the CQA test set and 93.4% accuracy on the dev set - however, it relied on access to ConceptNet, on which CQA is built.

**Case Study**  Note that it is harder to judge the rationale quality: for arithmetic, one can compare them to the ground truth rationales, but for CQA the evaluation is necessarily qualitative. For this reason, we include a case study in Figure 7. We observe that the rationales provided are generally coherent and of a similar structure to the few-shot rationales. We make the following two observations:

1. After training with STaR, we see the model was able to generate reasonable rationales that solve new problems, which explains part of the observed performance gain.

2. We also see that there were many instances in which STaR improved the quality of rationales over those generated in a few-shot manner.

**Human Evaluation**  Based on the observation that STaR may improve reasoning quality for problems even when they were initially answered correctly via few-shot prompting, we performed a preliminary qualitative analysis. We randomly selected 50 rationales generated from few-shot CoT and STaR-generated rationales on questions which they both answered correctly, as well as human-generated rationales for these problems from [3]. We then presented a random subset of 10 questions and rationales to each of 20 crowdworkers on Prolific [34] with the rationales in a randomized order, asking them to rank the rationales based on which they felt best justified the answer. The participants were 30% more likely to rank the STaR-generated rationales higher than the few-shot rationales ($p = .039$). This indicates that, as mentioned in the case study, STaR can improve the quality of rationale generation.

---

[4]We use the same few-shot rationales as described in Section 4.1 - namely fixing typos and improving clarity.

Table 2: We find that STaR substantially improves GSM8K performance over the baselines, despite training on only 25.0% of the data for the model without rationalization, and 30.3% of the dataset (with 1.2% from rationalization) for the model with rationalization.

|  | GSM8K Test Accuracy (%) | Train Data Used (%) |
|---|---|---|
| Few-shot Direct GPT-J | 3.0 | ∼0 |
| Few-shot CoT GPT-J | 3.1 | ∼0 |
| GPT-J Direct Finetuned | 5.8 | 100 |
| STaR without rationalization | 10.1 | 25.0 |
| STaR with rationalization | **10.7** | 30.3 |

We also found that the participants were 74% more likely to prefer the STaR-generated rationales over the human-generated rationales ($p < .001$). To be clear, we do not believe that this indicates human-level rationale-generation performance. Instead, we feel that it speaks to the difficulty of eliciting high-quality rationales. We reproduce the test prompts in Appendix C and elaborate on the limitations of the crowdsourced explanations dataset.

**Failure Cases**   Finally, we found a variety of interesting failure cases, many of which corresponded to standard logical fallacies. For example, the model often made statements related to the topic of the question but which were not actually arguments for why the answer should be true. Sometimes, the model claimed the question implied the answer as an argument, without explaining why. Other times, especially early in training, the model answered as if it has knowledge about a particular individual, instead of making a general statement - e.g. "the king's castle is a place where he feels safe" instead of "castles are places where kings feel safe." We provide examples and analyze errors in Appendix A.

**Few-shot Prompt Training**   Using few-shot prompts during fine-tuning [12] appears to have a meaningful benefit (60.9%→68.8% without rationalization, 69.9%→72.5% with rationalization). Thus, we generally suggest its use for at least a portion of training, though we discuss caveats in Section 5.

### 4.5   Mathematical Reasoning in Language: Grade School Math

We again find on GSM8K that STaR substantially improves performance beyond few-shot with rationales or training to directly predict the answers (without rationales), shown in Table 2 and include the few-shot prompt in Appendix I. We observe that on this task, rationalization does not substantially improve performance. Note that, in training, it was necessary to cap the number of training steps at the 30th iterations (after 7912 steps), to prevent training from becoming prohibitively long. The results were reached after 36 iterations for STaR without rationalization and an additional 12 iterations with rationalization.

Usually, the number of calculation steps generated by the model matches the number of steps taken by humans (generally 53-57% agreement across all iterations). We visualize this explicitly in Figure 6. We see that when the ground truth and model disagree on the number of calculation steps, the model typically uses

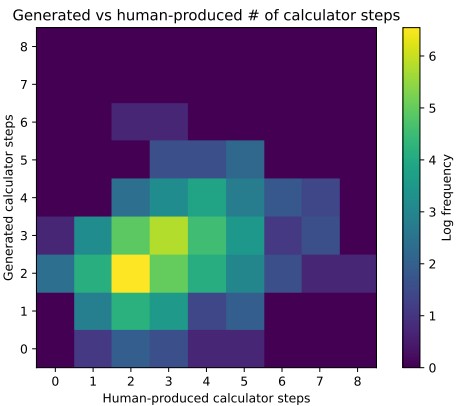

Figure 6: A comparison of the number of calculator steps generated by the model to solve examples in the training set relative to the number of steps used in the ground truth.

fewer. Sometimes this is because the model skips steps, but occasionally it finds novel solutions, as in Appendix J, where the model disregards redundant details and solves a 7-step problem in one step.

## 5   Discussion and Challenges

**The Impact of Rationalization**   An essential question is exactly what role rationalization plays. Intuitively, rationalization allows a model to reverse-engineer a solution, or provides a heuristic for identifying whether each step makes the conclusion more likely. This parallels real-world problems where the final result is known, but challenging to derive a good justification. From a mathematical perspective, while rationale generation samples rationales from the distribution $p(r \mid x)$ provided by our model $M$, rationalization conditions on the answer, letting us access an alternative distribution $p(r \mid x, y)$ which may be a better search space for rationales. Then rationalization could be framed as an off-policy estimate of the objective in Equation 1, sampling from the hint-augmented model as a

proposal distribution. Future work should establish more connections between rationalization and these RL objectives, and examine more generally when and why rationalization improves learning.

In addition, due to the low sampling temperature, outputs without rationalization correspond to examples where the model is most confident in its answer. This results in these examples providing a weaker gradient signal than the rationalization examples, at least in the first iteration. Since we retrain from the initial pre-trained model every time we run a fine-tuning iteration, the degree of this effect is also difficult to measure directly. Finally, we must point out that the method to add the "hint" does not follow immediately from the question and answer and in some contexts providing it may be nontrivial. Exploring the impacts of different hinting techniques and their generality is an avenue for future work.

**Temperature**   One intuitive alternative to rationalization, if one seeks to expand the training dataset, is more and higher-temperature sampling. However, in practice, we found that this is counterproductive. In general, it substantially increases the likelihood of a correct answer despite incorrect reasoning, and training on bad or irrelevant reasoning prevents generalization. This is particularly clear in more structured tasks, like arithmetic, where the scratchpads that the model learns to produce with a higher-temperature sampling approach diverge into meaninglessness and cause the model to stagnate. Overall, we found that higher temperatures as an alternative to rationalization (e.g. $0.5$ or $0.7$) led to models worse than models with reasoning alone, discussed further in Appendix H.

Furthermore, as text generation by large language models is sequential (i.e. one cannot produce a token without producing the preceding token), generating text is a bottleneck and this is computationally far less efficient than rationalization. For example, generating 10 sample outputs is approximately 10 times slower than generating one sample output. However, one potentially valuable way to leverage multiple samples would be to use the method proposed in [35], using the majority-vote result of multiple high-temperature scratchpads as a ground truth against which we compare a low-temperature scratchpad. This allows one to apply STaR to a dataset of only questions, without answers. This significantly underperformed using ground truth, but we discuss this ablation in Appendix H.

**Few-shot Prompting**   A noteworthy phenomenon is that the inclusion of few-shot prompting during sampling seems to dramatically reduce "drift" where later rationales become increasingly dissimilar from the initial few-shot set of rationales. One benefit of this is that the model may be less constrained by the quality and difficulty of the initial rationales, theoretically allowing it to generalize more. One potentially negative consequence is that the style of the rationales may less-closely match the original prompting style. Another benefit is in terms of computational resources - a shorter prompt length allows for a shorter sequence length when sampling. Technically, the point in training at which we "disable" few-shot prompts is another hyperparameter which we could tune, but we leave this to future work. In addition, by leaving prompts out after the initial outer-loop iteration, the model tends to perform gradually worse at rationalization as it trains for longer periods of time. As a result, it may be necessary to include some hints during training for long periods of time with this approach.

Ultimately, the choice to include few-shot prompts in later iterations of training appears to depend on the use-case: when the goal is consistent adherence to a particular prompt style, which may benefit explainability, include few-shot prompts in sampling; when the goal is a faster training loop, one may remove them. Moreover, it is possible that with other datasets or larger models there is an impact on performance, so we encourage this to be generally treated as a hyperparameter.

**Incorrect Rationales with Correct Answers**   One limitation of STaR is that undesirable rationales (e.g. for the reasons discussed in Appendix A or later in this section) paired with correct answers will still be used for training. These examples may detract from the performance simply by providing unclean training data. In addition, it is possible that the ultimate performance of the model may be worsened by training on rationales which are useful in a limited context but fail to generalize, relative to training with non-generalizable rationales filtered out. How to identify correct and generalizable rationales beyond checking the final answer (e.g. by using token-level verifiers as in [9]) is a valuable direction for future study. Still, in real-world contexts there are significant consequences to user trust and risk from a model producing bad or unfaithful explanations [36], discussed also in Appendix 6.

## 6   Limitations and Impacts

**Bias**   It is important to note that STaR is designed to amplify the reasoning leading to correct solutions on a dataset. Another implication is that if biases are "useful" in solving a dataset then they will be amplified. This is made worse by rationalization, as biased answers that the model may not

naturally arrive at are, in a sense, pulled out of the model. The exact interaction between the bias in a dataset and the pre-existing bias learned by the model is unclear, and something to consider before real-world deployment of all large language models in general, but STaR-trained models in particular.

We find some encouraging initial results on this however: for questions where gender is not relevant, the model appears to disregard it in its explanation. For example: "Q: Where is a good place for a woman to store her sunglasses? → A: The answer must be a place where sunglasses are stored. Sunglasses are stored in purses. Therefore, the answer is purse (e)." or "Q: The only baggage the woman checked was a drawstring bag, where was she heading with it? → A: The answer must be a place where a drawstring bag is checked. The answer is airport (e)." We believe this question warrants a much more comprehensive study.

**Faithfulness**    One important challenge with models which seek to be interpretable or provide explanations for their reasoning is that of faithfulness. While our primary emphasis is not on the explainability benefits that STaR may bring, there is a fundamental challenge around evaluating explanations and rationales: namely, faithfulness [37, 38]. [38] describe faithful explanations as those which "accurately [represent] the reasoning process behind the model's prediction." While STaR encourages the use of reasoning in rationales which leads the model to correct answers, it is difficult, if not impossible, to ensure that the rationales reflect the model's internal processing. For example, it is straightforward to imagine the model implicitly selecting a particular answer immediately and then generating a rationale to justify that selected answer. This would allow a model to generate unbiased rationales while selecting answers in a biased way.

The fact that our model outperforms one fine-tuned to directly predict the answers, and ablation studies from papers such as [6] make it clear that the generation of a rationale before producing an answer non-trivially improves the model's answer quality. However, it is difficult to evaluate the degree to which any particular answer's rationale is faithful. However, we note that there problems are not unique to STaR, but are symptomatic of the difficulty of understanding large language models and in particular the rationales generated by large language models.

**Scale**    Finally, we note there is no guarantee that our results would generalize to larger models. However, [39] and [40] suggest that the benefits of rationales increases with scale on numerous problems, which is perhaps reason for optimism. On the other hand, a limitation of STaR on small models is that in order for the first iteration of STaR to succeed, few-shot performance must be above chance. This implies that the initial model must be big enough to have some reasoning capabilities. For instance we found that GPT-2 was not able to bootstrap from few-shot reasoning in even the arithmetic domain. A further limitation is that settings with a high level of chance performance (e.g. binary decisions) yield many poor rationales, confounding the STaR approach. As discussed, filtering bad reasoning paired with correct answers remains an open question.

## 7    Conclusion

We present the Self-Taught Reasoner (STaR), which iteratively improves a model's ability to generate rationales to solve problems. We few-shot prompt a model to solve many problems in a step-by-step manner by generating rationales, and then prompt it to rationalize the correct answer for problems it gets wrong. We finetune on both the initially correct solutions and rationalized correct solutions, and repeat the process. We find that this technique significantly improves the model's generalization performance on both symbolic reasoning and natural language reasoning. There are several key limitations on STaR as discussed in Section 5 and Appendix 6. Nonetheless, we believe using examples without reasoning to bootstrap reasoning is a very general approach, and that STaR can serve as the basis of more sophisticated techniques across many domains.

## Acknowledgements

We thank Imanol Schlag for his detailed feedback about this work, as well as Rose E Wang, Markus Rabe, Aitor Lewkowycz, Rishi Bommasani, Allen Nie, Alex Tamkin, and Qian Huang. We thank Cem Anil for his very helpful insight that rationale finetuning performance can be improved if the training includes the few-shot rationales. We also thank Ben Prystawski for his suggestions on survey creation. We thank Google TPU Research Cloud for TPU access. This work was partially supported by SAIL, an Open Phil AI Fellowship (for JM), and an NSF Expeditions Grant, Award Number (FAIN) 1918771.

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
