# Appendix

## A  CommonsenseQA Error Patterns

Throughout our experiments, we came across a variety of interesting failure cases for commonsense reasoning. Note that all the final answers are correct – however, we take issue with the reasoning used in order to arrive at those answers.

### A.1  Question Implies Answer

One key failure case was answers in the form of "the answer must be something that is <question property>. <answer> is <question property>. Therefore, the correct answer is <answer>." In these cases, the model fails to explain *why* the answer that it has chosen satisfies the question property.

```
Q: What might someone get from learning about science?
Answer Choices:
(a) headache
(b) see things differently
(c) increased knowledge
(d) accidents
(e) appreciation of nature
A: The answer must be something that someone would get from learning about
science. Learning about science would increase knowledge. Therefore, the
answer is increased knowledge (c).
```

These rationales, while perhaps useful to the model, read to us as opaque and unexplanatory.

### A.2  Begging the Question

A related but stronger version of the previous failure case, while less common, is particularly uninsightful. Sometimes the model will imply the answer that it has chosen in its question.

```
Q: The christian rock band was on a large stage, the singer's foot got caught
on a wire and he had a small what?
Answer Choices:
(a) commit sin
(b) fail
(c) disobey
(d) stumble
(e) boo boo
A: The answer must be something that would cause the singer to stumble. The
singer's foot got caught on a wire, so he would stumble. Therefore, the
answer is stumble (d).
```

### A.3  Exercise to the Reader

A rare failure case is when the model finds it unnecessary to justify its answer. For example:

```
Q: Other than school or from text where can one gain knowledge?
Answer Choices:
(a) book
(b) class
(c) thinking
(d) head
(e) meeting
A: The answer must be a place where someone can gain knowledge other than
school or from text. The answer is meeting (e).
```

## A.4 World State Assertions

Sometimes, the model will assume that it knows something about a subject or a person whose name was used as a variable. This leads to somewhat comical examples of reasoning. Part of the reason for this is that generally, there is an expectation that good rationales will leverage understanding of more general classes of objects and appeal to the relationship between those general classes and the particular instance. For example, the argument that "a person would typically feel exhilaration from heights" is generally more compelling than the argument that "James would feel exhilaration from heights."

```
Q: A person's hand hurts after hitting the bottle. What might happen to
the bottle?
Answer Choices:
(a) alcoholism
(b) breaking
(c) fly
(d) nausea
(e) ticket
A: The answer must be something that happens to the bottle. The bottle is
broken. Therefore, the answer is breaking (b).

Q: James loved climbing.  What emotion might he feel from heights?
Answer Choices:
(a) exhaustion
(b) exhilaration
(c) falling down
(d) nervousness
(e) vertigo
A: The answer must be a feeling that would cause someone to climb a mountain.
James would feel exhilaration from heights. Therefore, the answer is
exhilaration (b).
```

## A.5 Red Herrings

Some errors in reasoning corresponded to the model making a statement which, while technically true, is not useful in demonstrating the claim.

```
Q: A squirrel ran up to it's home, where is the small dog likely to be
barking up?
Answer Choices:
(a) manhattan apartment
(b) wood
(c) neighbor's yard
(d) tree
(e) basket
A: The answer must be a place where a small dog would be likely to be
barking. Squirrels are not small dogs. Therefore, the answer is tree (d).

Q: John ran very fast, his feet touched the ground in what way?
Answer Choices:
(a) behind
(b) loose
(c) feverishly
(d) slowly
(e) lightly
A: The answer must be a way of running that is different from the way John
ran. Running is a fast, light-footed activity. Therefore, the answer is
lightly (e).
```

## A.6 Hint Short-cutting

In the experiments where the model was shown some examples of "hints" during training, in order to prevent it from losing the ability to perform rationalization over time, the model appeared to pick up on the fact that the final answer would always correspond to the hinted answer. This led to answers such as

```
Q: Billy bought coffee and waited for his wife to arrive from  France.
Where might he have been?
Answer Choices:
(a) airport
(b) grocery store
(c) internet cafe
(d) supermarket
(e) train station
A: The answer must be a place where Billy could have been waiting for his
wife to arrive from France. The airport is a place where people can wait
for flights. Therefore, the answer is train station (e).
```

# B Modified CQA Prompts

For reference, we include our modified prompts based closely on those in [6].

Q: What do people use to absorb extra ink from a fountain pen?
Answer Choices:
(a) shirt pocket
(b) calligrapher's hand
(c) inkwell
(d) desk drawer
(e) blotter
A: The answer must be used to absorb extra ink. Blotters are designed to absorb
liquids. Therefore, the answer is blotter (e).

Q: What home entertainment equipment requires cable?
Answer Choices:
(a) radio shack
(b) substation
(c) television
(d) cabinet
(e) desk
A: The answer must require cable. Cable is used to provide satellite channels to
televisions. Therefore, the answer is television (c).

Q: The fox walked from the city into the forest, what was it looking for?
Answer Choices:
(a) pretty flowers
(b) hen house
(c) natural habitat
(d) storybook
(e) dense forest
A: The answer must be a reason for a fox to go into the forest. The forest is a fox'
s natural habitat. Therefore, the answer is natural habitat (c).

Q: Sammy wanted to go to where the people were. Where might he go?
Answer Choices:
(a) populated areas
(b) race track
(c) desert
(d) apartment
(e) roadblock
A: The answer must be a place with many people. Populated areas, by definition, have
 a lot of people. Therefore, the answer is populated areas (a).

Q: Where do you put your grapes just before checking out?
Answer Choices:
(a) mouth
(b) grocery cart
(c) super market
(d) fruit basket
(e) fruit market
A: The answer should be the place where grocery items are placed before checking out
. Of the above choices, grocery cart makes the most sense for holding grocery items.
 Therefore, the answer is grocery cart (b).

Q: Google Maps and other highway and street GPS services have replaced what?
Answer Choices:
(a) united states
(b) mexico
(c) countryside
(d) atlas
(e) oceans
A: The answer must be something that used to do what Google Maps and GPS services do
, which is give directions. Atlases were also used to give directions. Therefore,
the answer is atlas (d).

Q: Before getting a divorce, what did the wife feel who was doing all the work?
Answer Choices:
(a) harder
(b) anguish
(c) bitterness
(d) tears
(e) sadness
A: The answer should be a feeling which would cause someone who was doing all the work to get divorced. If someone feels bitter towards their spouse, they are likely to want a divorce. Therefore, the answer is bitterness (c).

## C   Human-evaluated Test Prompts

We also selected a random sampling of 50 questions which were correctly answered both few-shot and by a STaR-trained model (without rationalization), as discussed in 4.4. Presented in a random order, twenty crowdworkers preferred the STaR-generated answers. We reproduce the examples here with the few-shot rationale first, the STaR-trained rationale second, and the human rationale third, though these were shuffled when presented to participants. We selected human answers from [3]'s original split rationales where possible, finding that duplicate rationales were much more common in the new split rationales. For example, the explanation "Rivers flow trough valleys," appeared over 400 times verbatim in the new split dataset, and "This word was most relevant" appeared over 150 times. [3]'s dataset also includes explanations like "The only answer that makes sense" or "BOB WILL NOT POKEMON CARDS WERE COMMON AND WORTHLESS BUT WRONG ABOUT THEM SO FEEL REALLY RARE TO DELAY" or restatements of the answer. We append the phrase "Therefore, the answer is ANSWERTEXT (ANSWERLETTER)" with ANSWERTEXT replaced by the correct answer's text and ANSWERLETTER replaced by the correct answer letter. This is done 1) to make it less obvious that one of the answers is generated by a different source and 2) to prioritize differences in rationales, not the answer format.

Before the questions and after the consent form, we presented the following directions:

```
Each question will have three explanations for one answer, which is a pre-determined correct ("best") answer.

Please rank the explanations for each question from best to worst (with 1 as the best and 3 as the worst) based on how well
they justify the correct answer.

Note there are some typos and grammatical errors in the examples, in the questions, answers, and the explanation. Please
treat the answer letter (a / b / c / d / e) at the end of the explanations as the correct answer letter, even if another
answer is valid (and perhaps more reasonable), and even if it does not match with the answer text preceding it.
```

The examples were subsampled and presented to the crowdworkers:

```
Q: Who is likely yo have a caring heart?
Answer Choices:
(a) person
(b) species
(c) artichoke
(d) all mammals
(e) live animal
 A: The answer must be a person. All mammals have a caring
 heart. Therefore, the answer is person (a).
 A: The answer must be someone who is likely to have a
 caring heart. Humans are the most likely to have a caring
 heart. Therefore, the answer is person (a).
 A: A person is the only human being capable of having a
 caring heart. Therefore, the answer is person (a).

Q: If a person wanted to become a good writer what should
they do in college?
Answer Choices:
(a) word sentence
(b) own animal
(c) read the newspaper
(d) catch cold
(e) study literature
 A: The answer must be something that would help a person
 become a good writer. The answer is study literature (e).
 A: The answer must be something that would help a person
 become a good writer. Literature is a form of study.
 Therefore, the answer is study literature (e).
 A: studying literature will give one skills of writing.
 Therefore, the answer is study literature (e).

Q: The human always would have fun making up questions for
the A.I. overlords, he found the task quite what?
Answer Choices:
(a) do enjoy
(b) eat cake
(c) enjoy living
(d) get laid
(e) enjoyable

 A: The answer must be something that the human would enjoy
 doing. The human enjoys making up questions for the A.I.
 overlords. Therefore, the answer is enjoyable (e).
 A: The answer must be something that the human found
 enjoyable. The human found the task of making up questions
 for the A.I. overlords enjoyable. Therefore, the answer is
 enjoyable (e).
```

```
 A: Having fun is enjoyable. Therefore, the answer is
 enjoyable (e).

Q: What happens when I make a pig of myself eating breakfast
?
Answer Choices:
(a) full stomach
(b) gain energy
(c) dieting
(d) have energy
(e) feeling satisfied

 A: The answer must be a feeling which would cause someone
 to feel full after eating breakfast. The answer is full
 stomach (a).
 A: The answer must be a feeling which would cause someone
 to make a pig of themselves eating breakfast. If someone
 feels full, they are likely to stop eating. Therefore, the
 answer is full stomach (a).
 A: A colloquial phrase for over indulging in food.
 Therefore, the answer is full stomach (a).

Q: In order to see a story on the big screen what must you
do?
Answer Choices:
(a) go to movies
(b) visualize
(c) reading
(d) open book
(e) sketching a picture

 A: The answer must be something that you do to see a story
 on the big screen. Movies are a way to see a story on the
 big screen. Therefore, the answer is movies (a).
 A: The answer must be something that is required to see a
 story on the big screen. Movies are a form of storytelling.
  Therefore, the answer is go to movies (a).
 A: The big screen is a colloquial term meaning a movie
 theater. Therefore, the answer is go to movies (a).

Q: What is the opposite of remembering?
Answer Choices:
(a) answering question
(b) stopping to think
(c) knowing
(d) closing eyes
```

(e) forgetting
A: The answer must be something that is the opposite of remembering. The answer is forgetting (e).
A: The answer must be a feeling which would cause someone to not remember something. If someone forgets something, they are likely to not remember it. Therefore, the answer is forgetting (e).
A: Health complications. Therefore, the answer is forgetting (e).

Q: If you aren't experiencing curiosity and want to ensure viewing enjoyment, what may you watch?
Answer Choices:
(a) examine thing
(b) stare into a mirror
(c) go somewhere
(d) hear news
(e) see favorite show
A: The answer must be something that will make you want to watch. If you are not experiencing curiosity, you may want to watch something that will make you want to watch. Therefore, the answer is watch (e).
A: The answer must be something that will not cause you to want to watch. If you are not experiencing curiosity, you are likely to want to watch something that is entertaining. Therefore, the answer is favorite show (e).
A: see favorite show can you do in front of a tv if you have curiosity about what is being displayed on the tv. Therefore, the answer is favorite show (e).

Q: The carotid is a main artery susceptible to damage where?
Answer Choices:
(a) neck
(b) face
(c) busy city
(d) torso
(e) human body
A: The answer must be a place where the carotid artery is susceptible to damage. The carotid artery is located in the neck. Therefore, the answer is neck (a).
A: The answer must be a place where the carotid is susceptible to damage. The carotid is susceptible to damage in the neck. Therefore, the answer is neck (a).
A: The carotid artery is located in the neck. Therefore, the answer is neck (a).

Q: The king needed to feel safe, where did he go?
Answer Choices:
(a) castle
(b) throne room
(c) deck of cards
(d) fort
(e) court
A: The answer must be a place where the king feels safe. The king's castle is a place where he feels safe. Therefore, the answer is castle (a).
A: The answer must be a place where the king can feel safe. Castles are places where kings feel safe. Therefore, the answer is castle (a).
A: In general, King live in castle. Therefore, the answer is castle (a).

Q: What is the result of instituting civil action?
Answer Choices:
(a) bad feelings
(b) confusion
(c) going to court
(d) hurt feelings
(e) sandwiches
A: The answer must be a result of instituting civil action. The result of instituting civil action is a lawsuit. Therefore, the answer is going to court (c).
A: The answer must be a result of instituting civil action. If someone goes to court, they are likely to feel hurt feelings. Therefore, the answer is going to court (c).
A: Anger and Trial will not institute a court case. Therefore, the answer is going to court (c).

Q: The person wanted to fix his car, what did he order?
Answer Choices:
(a) selling things
(b) gas
(c) feel resentful
(d) machine part
(e) cross street
A: The answer must be something that would help the person fix his car. The person would not want to sell things, because he would not want to give up his car. Therefore, the answer is machine part (d).
A: The answer must be something that would be used to fix a car. Machine parts are used to fix cars. Therefore, the answer is machine part (d).

A: A machine part is the only thing listed that would fix the car. Therefore, the answer is machine part (d).

Q: He always traveled up the mountain when winter arrived, he had a passion and desire to do what?
Answer Choices:
(a) play volleyball
(b) play hockey
(c) ski
(d) knit
(e) skate
A: The answer must be something that the person does when winter arrives. The answer is skiing (c).
A: The answer must be something that the person did when winter arrived. The answer is ski (c).
A: Skiing is the only option that is done on a mountain. Therefore, the answer is ski (c).

Q: Bob thought that his Pokemon cards were common and worthless, but he was wrong about them. They were really what?
Answer Choices:
(a) personal
(b) special
(c) shiny
(d) rare
(e) irregular
A: The answer must be something that is not common and worthless. Pokemon cards are not common and worthless. Therefore, the answer is rare (d).
A: The answer must be something that are rare. Pokemon cards are rare. Therefore, the answer is rare (d).
A: BOB WILL NOT POKEMON CARDS WERE COMMON AND WORTHLESS BUT WRONG ABOUT THEM SO FEEL REALLY RARE TO DELAY. Therefore, the answer is rare (d).

Q: If you're lighting a candle to see what are you in?
Answer Choices:
(a) dimly lit room
(b) synagogue
(c) sunlight
(d) birthday cake
(e) bedroom
A: The answer must be a place where you can see what you are in. The answer is a room with dim lighting. Therefore, the answer is dimly lit room (a).
A: The answer must be a place where you can see a candle. Candles are used to see things in the dark. Therefore, the answer is dimly lit room (a).
A: Lighting or illumination is the deliberate use of light to achieve a practical or aesthetic effect. Lighting includes the use of both artificial light sources like lamps ... Therefore, the answer is dimly lit room (a).

Q: The person was thirsty, what should he have done?
Answer Choices:
(a) eat
(b) sunny days
(c) compliments
(d) drink water
(e) all kinds of things
A: The answer must be something that would make the person feel better. If the person is thirsty, he should drink water. Therefore, the answer is drink water (d).
A: The answer must be something that would make someone feel better if they were thirsty. If someone is thirsty, they are likely to want to drink water. Therefore, the answer is drink water (d).
A: The person was thirsty, so drink water. Therefore, the answer is drink water (d).

Q: Where would you find many products for sale?
Answer Choices:
(a) market
(b) cost one dollar
(c) shelf
(d) salon
(e) demonstration
A: The answer must be a place where many products are sold. A market is a place where many products are sold. Therefore, the answer is market (a).
A: The answer must be a place with a lot of products for sale. Markets are the most common place to buy products. Therefore, the answer is market (a).
A: The only place products are for sale are in markets and stores. Therefore, the answer is market (a).

Q: If man and mammoth wanted to meet, they would probably have to do it during when?
Answer Choices:
(a) a baby shower
(b) ancient history

(c) storybook
(d) stone age
(e) forest
A: The answer must be a time when man and mammoth would
meet. Stone age is the time when man and mammoth lived.
Therefore, the answer is stone age (d).
A: The answer must be a time when man and mammoth would be
able to meet. The stone age was a time when man and mammoth
would be able to meet. Therefore, the answer is stone age
(d).
A: During the stone age is the only time mammoths lived.
Therefore, the answer is stone age (d).

Q: They prided themselves on being a wealth of knowledge,
and that's why many chose to attend their what?
Answer Choices:
(a) book store
(b) university
(c) meeting
(d) class
(e) encyclopedia
A: The answer must be a place where people go to learn
about things. The answer is university (b).
A: The answer must be a place where people go to learn
about things. Universities are places where people go to
learn about things. Therefore, the answer is university (b).

A: because in the university you get more knowledge.
Therefore, the answer is university (b).

Q: what happens when visiting museum that have narrow halls?
Answer Choices:
(a) sore feet
(b) amazement
(c) retrospection
(d) boredom
(e) crowding
A: The answer must be something that happens when visiting
a museum that has narrow halls. Crowding is a common
problem in museums. Therefore, the answer is crowding (e).
A: The answer must be something that would cause someone to
have sore feet if they are walking in narrow halls. If
someone feels cramped, they are likely to want to go
somewhere else. Therefore, the answer is crowding (e).
A: museum is one of the crowdest place. Therefore, the
answer is crowding (e).

Q: On their hike they brought a filtering straw, they were
worried about germs in the what?
Answer Choices:
(a) make sick
(b) doctor
(c) water
(d) stream
(e) mouth
A: The answer must be something that can filter out germs.
Filtering straws are used to filter out germs. Therefore,
the answer is filtering straw (c).
A: The answer must be something that would cause someone to
bring a filtering straw on a hike. Filtering straws are
used to filter water. Therefore, the answer is water (c).
A: straw it something used to drink water. Therefore, the
answer is water (c).

Q: If I am not good at playing poker, I will begin what if I
play?
Answer Choices:
(a) discord
(b) social event
(c) losing money
(d) competition
(e) heightened awareness
A: The answer must be something that will happen if you are
not good at playing poker. If you are not good at playing
poker, you will begin to lose money. Therefore, the answer
is losing money (c).
A: The answer must be something that will help you if you
are not good at playing poker. If you are losing money, you
are likely to want to stop playing. Therefore, the answer
is losing money (c).
A: Poker is a game in which people lose their money.
Therefore, the answer is losing money (c).

Q: People are meeting people, what is the likely physical
gesture?
Answer Choices:
(a) introducing yourself
(b) smiling
(c) shake hands
(d) share a hug
(e) talking

A: The answer must be a physical gesture that is used to
greet someone. The answer is handshake (c).
A: The answer must be a gesture used to meet people. If
someone wants to meet people, they are likely to want to
shake hands. Therefore, the answer is shaking hands (c).
A: A handshake is a globally widespread. Therefore, the
answer is shaking hands (c).

Q: What do dogs get that make them itch?
Answer Choices:
(a) bones
(b) fleas
(c) bad breath
(d) wagging tails
(e) four legs
A: The answer must be something that makes dogs itch. Dogs
get fleas, which make them itch. Therefore, the answer is
fleas (b).
A: The answer must be something that dogs get that make
them itch. Fleas are a common cause of dog itch. Therefore,
the answer is fleas (b).
A: fleas makes you itch. Therefore, the answer is fleas (b).

Q: What is ignorance often equated with?
Answer Choices:
(a) bliss
(b) hostility
(c) peace
(d) accidents
(e) find truth
A: The answer must be something that is often equated with
ignorance. Bliss is often equated with ignorance. Therefore,
the answer is bliss (a).
A: The answer must be a feeling which would cause someone
to be ignorant. If someone feels bliss, they are likely to
be ignorant. Therefore, the answer is bliss (a).
A: bliss comes from ignorance. Therefore, the answer is
bliss (a).

Q: The two teams were competing all the time, what likely
developed between them?
Answer Choices:
(a) accomplishment
(b) enemies
(c) rivalry
(d) friendship
(e) bad blood
A: The answer must be something that developed between the
two teams. The teams were competing all the time. Therefore,
the answer is rivalry (c).
A: The answer must be a feeling which would cause two teams
to compete all the time. Rivalry is a common feeling which
would cause two teams to compete all the time. Therefore,
the answer is rivalry (c).
A: likely developed between them of rivalry. Therefore, the
answer is rivalry (c).

Q: James doesn't want a relationship. He thinks that loving
another will only create what?
Answer Choices:
(a) satisfaction
(b) jealousy
(c) happiness
(d) wonderful
(e) quiet
A: The answer must be something that would make James feel
bad about not having a relationship. If someone feels
jealous, they are likely to want a relationship. Therefore,
the answer is jealousy (b).
A: The answer must be a feeling which would cause someone
who doesn't want a relationship to not want one. If someone
feels jealous of another, they are likely to not want a
relationship. Therefore, the answer is jealousy (b).
A: Jealousy is a negative emotion. Therefore, the answer is
jealousy (b).

Q: The had a reputation for celebrating in a debaucherous
manner, what often happened as a result?
Answer Choices:
(a) drunkenness
(b) enjoyment
(c) have fun
(d) hang over
(e) fighting
A: The answer must be a result of a celebration. The answer
is drunkenness (a).
A: The answer must be something that would cause someone to
have a reputation for celebrating in a debaucherous manner.
Alcohol is a depressant. Therefore, the answer is
drunkenness (a).

A: it is normal not a good idea to get drunk. Therefore,
the answer is drunkenness (a).

Q: The paper towels are usually in plain sight in the
kitchen, where did the dad go next to find them?
Answer Choices:
(a) pantry
(b) cabinet
(c) grocery store
(d) waste bin
(e) market
A: The answer must be a place where the dad would go to
find the paper towels. The pantry is a place where food is
stored. Therefore, the answer is pantry (a).
A: The answer must be a place where paper towels are stored.
 Pantries are the most likely place to store paper towels.
Therefore, the answer is pantry (a).
A: The pantry is in the kitchen and stores things.
Therefore, the answer is pantry (a).

Q: My house is very dry, with little water in the air. What
tool should I use to fix this?
Answer Choices:
(a) humidifier
(b) surface of earth
(c) skating rink
(d) ice cube
(e) burn it to the ground
A: The answer must be a tool that will make the air in your
 house moist. A humidifier is a tool that will make the air
 in your house moist. Therefore, the answer is humidifier (
a).
A: The answer must be something that would make the air in
my house more humid. A humidifier is used to make the air
in a room more humid. Therefore, the answer is humidifier (
a).
A: This word was most relevant. Therefore, the answer is
humidifier (a).

Q: A motorcycle is quicker because it isn't as what as a car
?
Answer Choices:
(a) heavy
(b) shiny
(c) red
(d) obesity
(e) expensive
A: The answer must be something that is not as heavy as a
car. A motorcycle is lighter than a car. Therefore, the
answer is not heavy (a).
A: The answer must be something that is not as quick as a
motorcycle. Motorcycles are not as heavy as cars. Therefore,
 the answer is heavy (a).
A: Because it is much smaller. Therefore, the answer is
heavy (a).

Q: The baseball team was playing ball. It was time for the
pitcher to act. How did the pitcher get the ball to the
batter?
Answer Choices:
(a) lose
(b) injury
(c) competition
(d) throwing
(e) rolled
A: The answer must be a way to get the ball to the batter.
The pitcher is the one who throws the ball. Therefore, the
answer is throwing (d).
A: The answer must be a way for the pitcher to get the ball
 to the batter. The pitcher is the person who is
responsible for getting the ball to the batter. Therefore,
the answer is throwing (d).
A: The pitcher threw the ball to the batter so that he
could hit it. Therefore, the answer is throwing (d).

Q: In what sort of container should you keep your ficus?
Answer Choices:
(a) refrigerator
(b) screened porch
(c) clay pot
(d) shady places
(e) arboretum
A: The answer must be a container that keeps the ficus in a
 shady place. Clay pots are used to keep plants in shady
places. Therefore, the answer is clay pot (c).
A: The answer must be a container in which ficus can be
kept. Ficus are tropical plants. Therefore, the answer is
clay pot (c).
A: Clay pot is the only type of container listed. Therefore,
 the answer is clay pot (c).

Q: James was cooking s stew in his apartment kitchen. Where
might he look for a potato.
Answer Choices:
(a) restaurants
(b) casserole
(c) root cellar
(d) pockets
(e) pantry
A: The answer must be a place where potatoes are stored.
The answer is pantry (e).
A: The answer must be a place where potatoes are stored.
Pantries are the most likely place to store potatoes.
Therefore, the answer is pantry (e).
A: apartments do not often have root cellars, and are not
restaurants. a pantry is closest to James inside his
apartment. Therefore, the answer is pantry (e).

Q: If ships are in a storm and the sun comes out, what has
happened?
Answer Choices:
(a) sinks
(b) cross river
(c) collide
(d) bottom out
(e) carry people
A: The answer must be a thing that happens when ships are
in a storm and the sun comes out. Ships are in a storm when
 they are in the middle of the ocean. Therefore, the answer
is bottom out (d).
A: The answer must be something that happens when the sun
comes out. The sun is a source of energy. Therefore, the
answer is bottom out (d).
A: the sun comes out so its not a storm any more. Therefore,
 the answer is bottom out (d).

Q: Where do all marmot live before they are captured?
Answer Choices:
(a) switzerland
(b) spain
(c) vancouver island
(d) wilderness
(e) australia
A: The answer must be a place where marmots live. Marmots
live in the wilderness. Therefore, the answer is wilderness
 (d).
A: The answer must be a place where marmots are found. The
wilderness is the most likely place to find marmots.
Therefore, the answer is wilderness (d).
A: Marmots don't live in Australia or Switzerland.
Therefore, the answer is wilderness (d).

Q: Where does one wait on a bench for something on tracks?
Answer Choices:
(a) train station
(b) state park
(c) crosswalk
(d) bus stop
(e) bus depot
A: The answer must be a place where one waits for a train,
bus, or other vehicle. The answer is train station (a).
A: The answer must be a place where one can wait for a
train. Train stations are places where trains stop.
Therefore, the answer is train station (a).
A: Only trains are on tracks. Therefore, the answer is
train station (a).

Q: What can planning vacation lead to between a couple when
they disagree?
Answer Choices:
(a) relaxation
(b) enjoying
(c) arguments
(d) going abroad
(e) spending money
A: The answer must be something that can happen when a
couple disagrees. Arguments are a common result of
disagreements. Therefore, the answer is arguments (c).
A: The answer must be a result of planning vacation.
Arguments are a common result of planning vacation.
Therefore, the answer is arguments (c).
A: Arguments are the direct result of disagreements.
Therefore, the answer is arguments (c).

Q: A sloth climbs slow, but it can eventually reach all the
way up where?
Answer Choices:
(a) forest canopy
(b) universe
(c) tropical rainforest
(d) large trees
(e) nature preserve

A: The answer must be a place where a sloth can climb. The forest canopy is the highest point in the forest. Therefore, the answer is forest canopy (a).
A: The answer must be a place where a sloth can climb. The canopy of a forest is the highest point in a forest. Therefore, the answer is canopy of a forest (a).
A: The sloth is a mammal that lives in the trees of South America. They have a slow and peculiar way of moving about. In fact, they are the slowest land mammals and travel very sluggishly on the ground (0.025 m/s-0.050 m/s). Therefore, the answer is canopy of a forest (a).

Q: Where in America is known for theaters?
Answer Choices:
(a) new york city
(b) shopping mall
(c) downtown area
(d) skating rink
(e) military base

A: The answer must be a place where people go to see movies. Theaters are places where people go to see movies. Therefore, the answer is theater (a).
A: The answer must be a place with a lot of theaters. New York City is known for its many theaters. Therefore, the answer is New York City (a).
A: New York City is the main region in US known for theaters. Therefore, the answer is New York City (a).

Q: What does hearing someone play violin beautifully make you?
Answer Choices:
(a) buy earplugs
(b) inspiring
(c) guitar
(d) make music
(e) like music
A: The answer must be something that makes you feel inspired. Hearing someone play violin beautifully makes you feel inspired. Therefore, the answer is inspiring (b).
A: The answer must be a feeling which would cause someone to want to hear someone play violin beautifully. Music is a form of art. Therefore, the answer is inspiring (b).
A: Hearing someone play a violin beautifully makes me inspired. Therefore, the answer is inspiring (b).

Q: What emotion does committing murder induce?
Answer Choices:
(a) fear
(b) go to jail
(c) problems
(d) cruel
(e) dead
A: The answer must be a feeling that would cause someone to commit murder. Murder is a crime. Therefore, the answer is fear (a).
A: The answer must be a feeling which would cause someone to commit murder. If someone feels fear, they are likely to want to avoid committing murder. Therefore, the answer is fear (a).
A: The only feeling. Therefore, the answer is fear (a).

Q: Sitting to close while watching TV can cause what sort of pain?
Answer Choices:
(a) brain problems
(b) laziness
(c) get fat
(d) headache
(e) laughter
A: The answer must be a pain that comes from sitting too close to the TV. Sitting too close to the TV can cause headaches. Therefore, the answer is headache (d).
A: The answer must be a type of pain which can occur if you are sitting too close to the TV. Headaches are a common side effect of sitting too close to the TV. Therefore, the answer is headache (d).
A: the process of aging in humans. Therefore, the answer is headache (d).

Q: What might people do to protect their legs from getting dirty on the farm?
Answer Choices:
(a) wear jeans
(b) milk cow
(c) believe in god
(d) avoid mud
(e) plant flowers
A: The answer must be something that people do to protect their legs from getting dirty. Of the above choices, wearing jeans is the most logical. Therefore, the answer is jeans (a).

A: The answer must be something that would protect legs from getting dirty on the farm. Jeans are used to protect legs from getting dirty. Therefore, the answer is wear jeans (a).
A: Jeans cover over legs. Therefore, the answer is wear jeans (a).

Q: What types of stoves are there?
Answer Choices:
(a) these twos are irrelevent to the questions
(b) warm pot of beans
(c) brown meat
(d) warm room
(e) gas or electric
A: The answer must be a type of stove. The stove is used to heat food. Therefore, the answer is gas or electric (e).
A: The answer must be a type of stove. Stoves are used to warm rooms. Therefore, the answer is electric (e).
A: stoves are gas or electric. Therefore, the answer is gas or electric (e).

Q: In what geological feature will you find fungus growing?
Answer Choices:
(a) shower stall
(b) toenails
(c) basement
(d) forrest
(e) cave
A: The answer must be a place where fungus grows. The answer is cave (e).
A: The answer must be a place where fungus grows. Fungus grows in caves. Therefore, the answer is cave (e).
A: Rivers flow trough valleys. Therefore, the answer is cave (e).

Q: What is our sun considered to be the center of?
Answer Choices:
(a) universe
(b) solar system
(c) outer space
(d) star
(e) beans
A: The answer must be something that is considered to be the center of the universe. The sun is the center of the solar system. Therefore, the answer is solar system (b).
A: The answer must be a place where the sun is the center of. The sun is the center of the solar system. Therefore, the answer is solar system (b).
A: It is the best option because it is the closest. Therefore, the answer is solar system (b).

Q: What are people in a library likely doing?
Answer Choices:
(a) talk to each other
(b) board ships
(c) study books
(d) suffer hunger
(e) playing games
A: The answer must be something that people in a library are likely to be doing. People in a library are likely to be studying books. Therefore, the answer is study books (c).

A: The answer must be something that is likely to happen in a library. Libraries are places where people study books. Therefore, the answer is studying books (c).
A: Books shelved as study-skills: How to Become a Straight-A Student by Cal Newport, The Study Skills Handbook by Stella Cottrell, Make It Stick: The Scienc... Therefore, the answer is study books (c).

Q: A shark isn't something you have to worry about at the lake in the woods, but there are plenty of dangers in the what?
Answer Choices:
(a) great oceans
(b) great outdoors
(c) tropical waters
(d) pacific ocean
(e) gulf of mexico
A: The answer must be something that is dangerous in the great outdoors. The great outdoors is a place where there are many dangers. Therefore, the answer is great outdoors (b).
A: The answer must be a place where sharks are a danger. Lakes in the woods are not a place where sharks are a danger. Therefore, the answer is great outdoors (b).
A: The great outdoors is the only choice where you would find a lake in the woods. Therefore, the answer is great outdoors (b).

Q: You can share files with someone if you have a connection to a what?

Answer Choices:
(a) freeway
(b) radio
(c) wires
(d) computer network
(e) electrical circuit
A: The answer must be a way to share files. The answer is computer network (d).
A: The answer must be a connection which allows you to share files. A computer network is a collection of computers which are connected to each other. Therefore, the answer is computer network (d).
A: Files can be transmitted over a computer network, and not a freeway. "wires" doesnt make any sense. Therefore, the answer is computer network (d).

Q: What can an injury happen to?
Answer Choices:
(a) emergency room
(b) doctor's office
(c) living thing
(d) bones
(e) accident
A: The answer must be something that can happen to a living thing. The answer is living thing (c).
A: The answer must be something that can happen to a living thing. If someone is injured, they are likely to want to go to the emergency room. Therefore, the answer is living thing (c).
A: Living things feel pain, and therefore can be injured. Therefore, the answer is living thing (c).

# D   Example Rationalizations on CQA

We include a randomly sampled set of rationalizations which the model is able to produce before fine-tuning. We observe that sometimes, the model constructs an argument roughly of the form "the answer must have *a set of properties*. *correct answer* has *those properties*. therefore, the answer is *correct answer*." This structure of argument is fairly standard, but given that the model originally answered those questions incorrectly, it resembles template-matching more than reasoning. The technique of rephrasing the question in a simpler way before answering it directly is also often effective. In general, the rationales from the pretrained model's rationalization appears to be of similar quality to the pretrained model's rationale generation. We reference the pretrained model here rather than the STaR-trained model, as the final iteration's rationales are the ones which the model struggled to learn from – that is, if the rationales were good, then we might expect that the model would have already incorporated them into its rationale generation.

```
Q: Why do I eat just enough when eating breakfast?
Answer Choices:
(a) gas
(b) full stomach
(c) feeling satisfied (CORRECT)
(d) have energy
(e) bloated
A: The answer must be a reason for eating just enough when eating breakfast. Eating
just enough is a way to feel satisfied. Therefore, the answer is feeling satisfied (
c).

Q: How is a dog likely to communicate with another dog?
Answer Choices:
(a) reproduce
(b) bark (CORRECT)
(c) do many things
(d) clone
(e) jump up
A: The answer must be a way for a dog to communicate with another dog. Dogs bark to
communicate with each other. Therefore, the answer is bark (b).

Q: If a person is trying to weasel out of punishment for a crime, where are they
likely to be?
Answer Choices:
(a) court room (CORRECT)
(b) cherry tree
(c) chicken coop
(d) natural history museum
(e) jail
A: The answer must be a place where a person would be trying to get out of
punishment for a crime. The answer is court room (a).

Q: What would easily hide a snake?
Answer Choices:
(a) living room of a house
(b) kew gardens
(c) terrarium
(d) thick forest (CORRECT)
(e) tropical forest
A: The answer must be something that would hide a snake. The answer is thick forest
(d).

Q: The person didn't like the varying size of the cake slices handed out, she demand
 more what?
Answer Choices:
(a) fairness (CORRECT)
(b) candies
(c) compliments
(d) well fed
(e) arguements
```

A: The answer must be a reason for the person to demand more cake. The person is not
 happy with the size of the cake slices. Therefore, the answer is fairness (a).

Q: In America, the courthouse of any given town has a likely location, where is it?
Answer Choices:
(a) electrical circuit
(b) getting married
(c) capital city
(d) center of town (CORRECT)
(e) michigan
A: The answer must be a location where a courthouse is located. The capital city is
the location of the capital of a state. Therefore, the answer is center of town (d).

Q: Where might I find a bowl nestled among other bowls in the kitchen?
Answer Choices:
(a) refrigerator
(b) kitchen top
(c) cupboard (CORRECT)
(d) dishwasher
(e) apartment
A: The answer must be a place where a bowl is found. The kitchen is a place where
bowls are found. Therefore, the answer is cupboard (c).

Q: Which state has blue ridge mountains that are called Blue Mountain?
Answer Choices:
(a) appalachians
(b) virginia
(c) pennsylvania (CORRECT)
(d) north carolina
(e) montana
A: The answer must be a state with mountains. The answer is Pennsylvania (c).

Q: Where could you find many radio studio?
Answer Choices:
(a) radio station
(b) country
(c) clear area
(d) office building
(e) large city (CORRECT)
A: The answer must be a place where many radio studios are located. Radio studios
are used to broadcast radio programs. Therefore, the answer is large city (e).

Q: Where would someone bring you a cup?
Answer Choices:
(a) apartment
(b) closet
(c) restaurant (CORRECT)
(d) table
(e) party
A: The answer must be a place where someone would bring you a cup. A restaurant is a
 place where people bring cups. Therefore, the answer is restaurant (c).

# E   STaR Without Rationalization (Rationale Generation Bootstrapping)

For convenience, we include the rationale generation bootstrapping algorithm alone here. However, it is simply Algorithm 1, with the blue components removed.

---

**Algorithm 2** Rationale Generation Bootstrapping (STaR without rationalization)

> **Input** $M$: a pretrained LLM; dataset $\mathcal{D} = \{(x_i, y_i)\}_{i=1}^{D}$ (w/ few-shot prompts)

1: $M_0 \leftarrow M$ # Copy the original model
2: **for** $n$ **in** $1...N$ **do** # Outer loop
3:    $(\hat{r}_i, \hat{y}_i) \leftarrow M_{n-1}(x_i) \quad \forall i \in [1, D]$ # Perform rationale generation
4:    $\mathcal{D}_n \leftarrow \{(x_i, \hat{r}_i, y_i) \mid i \in [1, D] \wedge \hat{y}_i = y_i\}$ # Filter rationales using ground truth answers
5:    $M_n \leftarrow \text{train}(M, \mathcal{D}_n)$ # Finetune the original model on the correct solutions - inner loop
6: **end for**

---

# F   Case Study

Q: John wants to eat at an Indian restaurant in on top of the space needle, where would he have to go?
Answer Choices:
(a) reservation (b) curry
(c) britsh town (d) seattle
(e) large city
A: The answer must be a place with a space needle. The space needle is a landmark in Seattle. Therefore, the answer is Seattle (d).

Q: I put a towel in a cardboard box for the kitten, I wanted it to have what?
Answer Choices:
(a) farmhouse (b) zoos (c) cold place
(d) floor (e) warm place
A: The answer must be a place where the kitten would be comfortable. The kitten would be comfortable in a warm place. Therefore, the answer is warm place (e).

Q: Needlepoint requires many accessories, what holds the work taut?
Answer Choices:
(a) sewing (b) collection of folk art
(c) livingroom (d) frame (e) bathroom
A: The answer must be something that holds the work taut. The work is taut because it is held in place. Therefore, the answer is frame (d).

(a) Problems where the model was able to solve questions which it had previously answered incorrectly.

Q: They prided themselves on being a wealth  of knowledge, and that's why many chose to attend their what?
Answer Choices:
(a) book store (b) university (c) meeting
(d) class (e) encyclopedia
~~A: The answer must be a place where people go to learn about things. The answer is university (b).~~
A: The answer must be a place where people go to learn about things. Universities are places where people go to learn about things. Therefore, the answer is university (b).

Q: You want a real estate agent that is straight forward with you, the last thing you want is for them to what a house's quality?
Answer Choices:
(a) cover up (b) offer price
(c) misrepresent (d) lie (e) falsify
~~A: The answer must be a way to determine the quality of a house. The answer is misrepresent (c).~~
A: The answer must be something that would be a bad thing for the real estate agent to do. The real estate agent would be misrepresenting the house's quality if they were to lie about it. Therefore, the answer is misrepresent (c).

(b) Problems where the model was already able to answer them with few-shot reasoning, but was able to answer better after more iterations.

Figure 7: Case study on STaR. We highlight some problems which the model is not initially able to solve, as well as rationales which are initially correct but improve. We note that for CQA, one is tasked with selecting the "best" answer, where multiple answers may be technically correct.

# G    Hyperparameters

GPT-J is a 28-layer decoder-only transformer, with an embedding size of 1024, 16 attention heads of dimension 256, and an FFN hidden layer of size 16384. It was pre-trained on the Pile [41], with a vocabulary size of 50.4K

In general, unless otherwise stated, we use a batch size of 8 sequences, each of length 1024. We also use packing, namely, packing the shorter examples to form longer sequences (up to length 1024) to improve TPU utilization. We do not use weight decay, and we train and sample on a single TPU-v3 node. We performed a hyperparameter search over learning rates from $10^{-7}$ to $10^{-4}$ using the Adam optimizer [42]. We found that $10^{-6}$ was consistently the best-performing learning rate.

# H    Further Analyses

## H.1    Ablations

We performed a set of additional experiments in order to validate the components of the method. Thanks again to the reviewers for their constructive and positive feedback.

1. Training with rationalization but without rationale generation on CQA reaches 69.2%, which is higher than rationale generation without rationalization (68.8%), but lower than the combination (72.5%).

2. Training on CQA without reinitializing the model reaches 70.6% accuracy.

3. Training with examples sampled with a temperature of of 0.5 on CQA reaches 72.0% accuracy, slightly worse than the main results.

4. In addition, some of the first ablations we performed studied the effects of removing ground truth in training. For example fine-tuning on the rationales without any filtering improved performance to 42.1% on CQA, and filtering by the majority vote led to 45.5%. Both were better than few-shot, but much worse than any fine-tuning with access to ground truth.

The drop from rationalization-only training is not unexpected, as it is presumably easier for the model to produce the right answer with a bad rationale if it is told the correct answer in advance.

## H.2    CQA Test Set

We also found that the model's performance on the CQA test set was 64.9%. This is about what one might expect, as the CQA test set is known to be more difficult than the dev set, with LLM in-context performance (e.g. GPT3) unavailable but other CQA models (e.g. T5, RoBERTa, XLNet, ALBERT, DREAM (ensemble)) losing about 5-8%.

## H.3    Iteration-wise Performance Dynamics

On the question of the stability/monotonicity of the improvement over iterations, the validation performance on CQA improves at almost every iteration, with at most a 1.2% decline relative to any point, and validation performance on GSM8K drops at most 0.8% from any iteration after improving past few-shot performance (3.1%).

On the cause of the difference between the arithmetic curves with and without rationalization. The accuracy for different lengths will improve at the same time to the extent that (some) longer problems are solved and added to the training data. We hypothesize that this happens more often when rationalization is used simply because it is easier to generate a scratchpad from the input numbers and final answer: all tokens in the scratchpad, aside from the carry token, are present in the prompt and the task is mostly to rearrange these tokens. This allows rationalization to generate some longer-length solutions earlier, which otherwise would not have been present. However, we feel that this could be a valuable question to explore more deeply in later work.

# I  GSM8K Few-shot Prompt

We include the following few-shot prompts for GSM8K, based on the examples in [9].

```
Q: Natalia sold clips to 48 of her friends in April, and then she sold half as many
clips in May. How many clips did Natalia sell altogether in April and May?
A: Natalia sold 48/2 = <<48/2=24>>24 clips in May.
Natalia sold 48+24 = <<48+24=72>>72 clips altogether in April and May.
#### 72

Q: Betty is saving money for a new wallet which costs $100. Betty has only half of
the money she needs. Her parents decided to give her $15 for that purpose, and her
grandparents twice as much as her parents. How much more money does Betty need to
buy the wallet?
A: In the beginning, Betty has only 100 / 2 = $<<100/2=50>>50.
Betty's grandparents gave her 15 * 2 = $<<15*2=30>>30.
This means, Betty needs 100 - 50 - 30 - 15 = $<<100-50-30-15=5>>5 more.
#### 5

Q: Julie is reading a 120-page book. Yesterday, she was able to read 12 pages and
today, she read twice as many pages as yesterday. If she wants to read half of the
remaining pages tomorrow, how many pages should she read?
A: Maila read 12 x 2 = <<12*2=24>>24 pages today.
So she was able to read a total of 12 + 24 = <<12+24=36>>36 pages since yesterday.
There are 120 - 36 = <<120-36=84>>84 pages left to be read.
Since she wants to read half of the remaining pages tomorrow, then she should read
84/2 = <<84/2=42>>42 pages.
#### 42

Q: Mark has a garden with flowers. He planted plants of three different colors in it
. Ten of them are yellow, and there are 80% more of those in purple. There are only
25% as many green flowers as there are yellow and purple flowers. How many flowers
does Mark have in his garden?
A: There are 80/100 * 10 = <<80/100*10=8>>8 more purple flowers than yellow flowers.
So in Mark's garden, there are 10 + 8 = <<10+8=18>>18 purple flowers.
Purple and yellow flowers sum up to 10 + 18 = <<10+18=28>>28 flowers.
That means in Mark's garden there are 25/100 * 28 = <<25/100*28=7>>7 green flowers.
So in total Mark has 28 + 7 = <<28+7=35>>35 plants in his garden.
#### 35

Q: Alexis is applying for a new job and bought a new set of business clothes to wear
 to the interview. She went to a department store with a budget of $200 and spent
$30 on a button-up shirt, $46 on suit pants, $38 on a suit coat, $11 on socks, and
$18 on a belt. She also purchased a pair of shoes, but lost the receipt for them.
She has $16 left from her budget. How much did Alexis pay for the shoes?
A: Let S be the amount Alexis paid for the shoes.
She spent S + 30 + 46 + 38 + 11 + 18 = S + <<+30+46+38+11+18=143>>143.
She used all but $16 of her budget, so S + 143 = 200 - 16 = 184.
Thus, Alexis paid S = 184 - 143 = $<<184-143=41>>41 for the shoes.
#### 41

Q: Tina makes $18.00 an hour. If she works more than 8 hours per shift, she is
eligible for overtime, which is paid by your hourly wage + 1/2 your hourly wage. If
she works 10 hours every day for 5 days, how much money does she make?
A: She works 8 hours a day for $18 per hour so she makes 8*18 = $
<<8*18=144.00>>144.00 per 8-hour shift
She works 10 hours a day and anything over 8 hours is eligible for overtime, so she
gets 10-8 = <<10-8=2>>2 hours of overtime
Overtime is calculated as time and a half so and she makes $18/hour so her overtime
pay is 18*.5 = $<<18*.5=9.00>>9.00
Her overtime pay is 18+9 = $<<18+9=27.00>>27.00
Her base pay is $144.00 per 8-hour shift and she works 5 days and makes 5 * $144 = $
<<144*5=720.00>>720.00
Her overtime pay is $27.00 per hour and she works 2 hours of overtime per day and
makes 27*2 = $<<27*2=54.00>>54.00 in overtime pay
```

2 hours of overtime pay for 5 days means she makes 54*5 = $270.00
In 5 days her base pay is $720.00 and she makes $270.00 in overtime pay so she makes
 $720 + $270 = $<<720+270=990.00>>990.00
#### 990

# J  STaR GSM8K Solutions

We observe some interesting patterns with the GSM8K solutions proposed by the STaR-trained model. Typically, when the solution takes substantially fewer calculation steps than the ground truth, it corresponds to an instance where the model accidentally answered the question correctly despite mistakes in its reasoning. In some cases, however, the model produces simpler solutions than those in the ground truth. One example is shown in Figure 8.

```
Q: A van is delivering  180 bottles of drinks to a neighborhood, and each bottle contains
either cider or beer or a mixture of two. Out of the  180 bottles, 40 contain only cider, 80
contain only beer, and the rest are a mixture of the two drinks. If the delivery man gives
half the number of each bottle of drink to the first house, how many bottles does the first
house get?
```

| Human (Ground truth from GSM8K dataset): | STaR (on GPT-J): |
|---|---|
| The total number of bottles containing cider or beer is 40+80 = <<40+80=120>>120 If the number of bottles in the van was 180, then 180-120 =<<180-120=60>>60 contains a mixture of the two drinks. At the first house, the delivery man dropped 1/2*40 = <<1/2*40=20>>20 bottles. There were also 1/2*80 = <<1/2*80=40>>40 bottles of beer that the delivery man gave to the first house. Together, the first house received 40+20 = <<40+20=60>>60 bottles of cider and beer The total number of bottles that had a mixture of the two drinks that were dropped at the first house is 1/2*60 = <<1/2*60=30>>30 The first house got 60+30 = <<60+30=90>>90 bottles of all types of drinks.

#### 90 | The delivery man gives half the number of each bottle of drink to the first house. So he gives 180/2 = <<180/2=90>>90 bottles of drink to the first house.

#### 90 |

Figure 8: An example problem in the training set where STaR derives a significantly simpler solution than the ground truth.