# OpenReview forum: "STaR: Bootstrapping Reasoning With Reasoning"
_NeurIPS.cc/2022/Conference — NeurIPS 2022 Accept_

### Official Review · Reviewer_UQiZ · 2022-07-03

**Rating:** 7
**Confidence:** 4
**Soundness:** 3 good
**Presentation:** 4 excellent
**Contribution:** 3 good

**Summary:**

This paper proposes a method for "bootstrapping reasoning" -- starting with a few examples of reasoning in a domain and producing supervised explanations for that domain. This uses a combination of two methods: generating an explanation and an answer and keeping the explanation if the answer is right, plus post-hoc rationalizing correct answers for cases where the model makes the wrong prediction. That is, the correct answer is used to "prime" the language model to generate an explanation geared towards that answer. Experiments are conducted on an arithmetic dataset, on GSM8k, and on CommonsenseQA.  Rationalization leads to faster learning for arithmetic, though not necessarily better "final" performance. On CommonsenseQA, the rationalized model does perform better and is able to use more of the training data. Human annotation shows that these generated explanations are of good quality.

**Questions:**

None

**Limitations:**

The limitations section was okay, although I feel like there are a few other chief limitations:

(1) Error propagation: if the model is bad at rationalizing, it is self-training on bad rationalizations. Even if they may somehow yield the right answer, there is a "right for the wrong reasons" concern here

(2) Potential non-generalizability of the results to larger models -- while possibly impractical for the authors to experiment with, the recent "emergent properties" paper (as well as discussion in GPT-3 and PaLM) shows that the fact of 10-100x in LM size makes a huge difference in results for tasks like these.

**Strengths And Weaknesses:**

STRENGTHS

- This is an exciting and very timely research direction. The core idea seems like a very good one and is likely to garner interest as interpretation generation in the context of these reasoning tasks becomes more widely studied.

- It's nice to see a diversity of problems tackled here: fully "rationalizable" (arithmetic) all the way to CommonsenseQA which has a very loose notion of what an explanation is.

- The experiments with GPT-J will be reproducible by broad swaths of the community with access to closed-off models, which is nice to see in this space.

- The discussion of the experiments in Section 5 is generally quite clear and helpful to shed light on the results

WEAKNESSES

- There's a pretty significant focus here on arithmetic, which in my opinion is not that interesting a problem to address with language models. It largely doesn't play to their strengths and is very synthetic in nature, though I admit it can be a useful test bed.

- Missing citations: While the application of this methodology to natural language reasoning / chain-of-thought is novel to my knowledge, this basic technique follows a long line of work on using weak supervision for problems like semantic parsing and knowledge base question answering. In these cases, the model does latent reasoning over possible derivations leading to the right denotation, which is essentially the same idea and has been described in the same policy gradient formulation before.

While the key idea dates back as far as Zettlemoyer and Collins's work, here are a few more recent citations:

Kelvin Guu et al. "From Language to Programs: Bridging Reinforcement Learning and Maximum Marginal Likelihood" https://arxiv.org/pdf/1704.07926.pdf

Omer Goldman et al. "Weakly Supervised Semantic Parsing with Abstract Examples" https://aclanthology.org/P18-1168.pdf

As a result, I don't think this paper is all that novel methodologically -- it's more applying an old idea to a new setting. This is fine as a contribution but more limited than what the paper claims.

---

> ### Comment · Reviewer_UQiZ · 2022-08-05
> **Thanks for the response**
>
> I appreciate your work to improve the paper!

---

### Official Review · Reviewer_932U · 2022-07-10

**Rating:** 6
**Confidence:** 4
**Soundness:** 2 fair
**Presentation:** 3 good
**Contribution:** 3 good

**Summary:**

This work introduces a boostrapping strategy for rationale generation to promote reasoning in question answering. Unlike existing generation methods, the boostrapping framework is able to iteratively improve rationale generations by filtering irrelevant rationales that lead to incorrect answers, and with another step of rationalization to revise the wrong rationales.

**Questions:**

(1) Comparing the rationale generation with and without rationalization, I start to wonder whether the authors have considered only using rationalization with hint to produce the rationales for fine-tuning. How does the performance differ from the proposed strategy? The reason for this question is that when filtering rationales leading to incorrect answers, it is possible that a good rationale is filtered, or a bad rationale is kept. Hence, it should be more reliable if providing the correct answer as hint to generate rationales.

**Limitations:**

yes.

**Strengths And Weaknesses:**

Strength:

The idea of bootstrapping rationale generation is interesting, and is specifically beneficial considering there is no annotated ground-truth rationales to learn from. Then it makes much sense to automatically filter those rationales that lead to incorrect answers and further augment the refined dataset with newly-generated rationales with another step of rationalization taking the correct answer as a hint. The authors conduct extensive experiments on three different domains to demonstrate the effect of bootstrapping along the iterations and the clear advantage in terms of performances over other baselines.

Weakness:

(1) The details for prompt design and format are not quite clear. Specifically, from Figure 2, it shows the prompt contains all the candidate choices, whereas there seems to be no choices in the prompt for rationale generation in section 3.1. How exactly does the format differ for rationale generation with and without rationalization? Does rationalization process also provides a number of demonstrations in its prompt?

(2) More experiments could be added: -- The current setting samples one rationale for each example. Since only one rationale is needed, I wonder how does greedy decoding perform (which brings less randomness). -- The authors mention that continually training the model may overfit and thus they re-train from $M_0$ for each iteration. It is better to empirically provide the evidence on how does continual training perform. -- Table 1 gives results on dev set of CQA. How about test set? -- Similar to Figure 4, do you evaluate the results on CQA along increasing iterations? Does the pattern align with Figure 4? -- For case study, it is better to provide some concrete examples to demonstrate the two points.

---

### Official Review · Reviewer_Ucno · 2022-07-12

**Rating:** 7
**Confidence:** 4
**Soundness:** 3 good
**Presentation:** 3 good
**Contribution:** 3 good

**Summary:**

This paper proposes a novel and interesting method for training pretrained large language models (LLMs) on reasoning tasks.
Specifically, an LLM is trained to output a rationale along with the answer.
If the predicted answer is correct, then the generated (question, rationale, answer) tuple is added to the training set, on which the model is further trained.
For questions that the model gets wrong, this paper proposes an additional “rationalization” step that trains the model to generate a new rationale by providing the correct answer.
This training approach allows the model to use its current reasoning capabilities to improve itself.
The model was applied to three diverse tasks - arithmetic, commonsense QA, and math-word problems.
Starting from only a few-shot dataset with rationales, the model self-generates a large training dataset with rationales and also performs few-shot models as well as models that are directly fine-tuned on the dataset (without rationales)

**Questions:**

It was not clear to me how you provide the hint to the model during rationalization. For example, for CommonSenseQA, what does the prompt look like with the hint? Is there a special marker on the correct answer?

In Figure 4a, the fact that the performance of (n+1)th digit starts to improve only after nth digit has been learned is quite clean and intuitive. However, with the model with rationalization, this behavior goes away, can you explain why?

Also, just to be clear in Fig 4a and b, the results on the validation sets are reported after each iteration, right?

What is the test accuracy on CommonsenseQA? Also please also report the state-of-the-art numbers on CommonSenseQA as presented in [29] (the same paper from where the GPT-3 numbers are taken from)

Minor: Line 178 - outer loop → inner loop

**Limitations:**

I believe it is important to explicitly include a section on the adverse effects of models generating flawed rationales while giving the right answer.

**Strengths And Weaknesses:**

Originality:

1. The paper presents a novel method for improving the performance of LLMs by using its current reasoning capabilities. It also leads to the generation of large scale rationale dataset, which is a bonus. While there are no novel technical ideas put forth by the paper, the general idea is novel and attractive and also generates interesting results

Quality:

2. The paper is generally of high quality with interesting ideas and well-done experiments on diverse benchmarks. Although, I think the paper should clearly report the current state-of-the-art numbers on CommonSenseQA.

Clarity:

3. The paper is very clearly written and was very easy to follow.

Significance:

4. Making LLMs generate its own rationale and use it for reasoning is a new research area with a lot of interest. I believe this paper brings in interesting results which will lead to further work in this area.

Weakness:

1. The biggest weakness of this approach is that there is no *guarantee* that the rationales generated by the model is correct even though it generates a right answer. Now this could easily be exacerbated as the model feeds these possibly flawed explanations to itself. The paper notes this limitation in a few places. But it is important to explicitly note this in a separate limitation section.

---

### Official Review · Reviewer_Gy9w · 2022-07-12

**Rating:** 7
**Confidence:** 3
**Soundness:** 3 good
**Presentation:** 4 excellent
**Contribution:** 3 good

**Summary:**

The paper builds on work on in-context learning and rationale generation.

The proposed approach uses a small seed set of query-answer pairs along with their rationales to prompt a large language model (GPT-J) to produce rationales along with the answer.

This prompted LM is then used to generate rationales for the remaining examples in the training set. When the predicted answer is correct, the generated rationale is added back into the training set to fine-tune the next version of the LM.

To prevent saturation where no new reasoning patterns are discovered, authors propose "rationalization". If the LM predicts the wrong answer, the correct answer is hinted to the model to elicit a new rationale and answer. If again, the answer is incorrect, the example is ignored. However, when given the hint, if the model predicts the right answer, then the associated rationale is added to the training set.

This iterative procedure of collecting more rationales and fine-tuning the LM on the larger dataset is repeated till saturation.

Authors show this procedure is useful to learn better performing models for simple arithmetic, common sense reasoning, and complex math problems; the model outperforms much larger models with fewer boot-strapped training examples. The generated rationales are analyzed for correctness and bias. An ablation is conducted to test the impact of "rationalization".

**Questions:**

Can authors provide concrete results backing claims made in Lines 354 and 371?

**Limitations:**

Appendix G contains a necessary discussion on the limitations of rationale generation systems and large language models more broadly. However, attention needs to be drawn to the section in the main body of the paper.

**Strengths And Weaknesses:**

Strengths
---
[+] The paper is clearly motivated, clearly written, and properly positioned with respect to past work.

[+] The method is intuitively explained along with possible connections to reinforcement learning.

[+] The rationalization procedure is ablated and described along with relevant caveats and warnings.

[+] Generated rationales are analyzed for correctness and bias.

Weakness
---
[-] Training stability is not discussed for all datasets. Only the instability in arithmetic reasoning is discussed.

[-] The number of cases of "right for the wrong reasons" is not estimated. There is a very good discussion on the types of errors, however, to paint a complete picture, there needs to be some estimation of what fraction of correct predictions were paired with incorrect rationales.

---

### Author Response · Authors · 2022-08-02
**Thank you for your questions and feedback!**

We would like to thank the reviewers for their thoughtful and encouraging comments, especially on the potential usefulness of STaR and on the paper's presentation. We also appreciate the supportive comments about reproducibility, dataset diversity, ablations, and exploratory analyses of biases.

As noted by Reviewers Gy9w, Ucno, and UQiZ, we strongly agree that STaR's limitations with respect to the impact of training on incorrect rationales should be given more attention in the main paper. We have added a paragraph to Section 5 discussing these limitations and will move the revised Appendix G on Further Limitations into the main text as Section 6 if the paper is accepted (as we are asked to keep to 9 pages for the rebuttal).

We also, as suggested by Reviewers Gy9w and 932U, are running ablations to properly quantify the negative impact of temperature sampling and continual training on CQA. As suggested by 932U we are also running a new rationalization-only ablation (i.e. STaR without first attempting to generate rationales from the question). We intend to follow up with the results of these ablations within a day. We will also incorporate these numerical results into the paper.

Reviewers Ucno and 932U asked about evaluations on the test set of CQA. We did not initially evaluate this since we do not have access to the answers in the dataset. However, we have now queried our model with the test set problems and emailed its predictions to the CQA team - we plan to update the paper with the test set evaluation numbers when they become available. We have also highlighted the overall SoTA on CQA from [29] in the text - though we noted that it relies on access to ConceptNet, from which CQA was built, and is a 39-model custom ensemble.
Based on questions about the prompt formats from Reviewers Ucno and 932U, we have changed to text to make it clearer that
* Figure 2 demonstrates a hint on CQA, and that for GSM8K we provided the hint as the final answer in parentheses immediately after the question.
* For the few-shot prompts for rationalization, we included the same questions but with the hint provided in the described format for those questions.
* We treat the answer choices, where provided, as part of the question/problem (i.e. $x_i$ in Section 3.1)

In addition, there were multiple questions about the training patterns from Reviewers 932U, Ucno, and Gy9w. Reviewers 932U and Gy9w asked whether the stability/monotonicity of the improvement in arithmetic also matches for the other datasets. The short answer is yes. More precisely, the validation performance on CQA improves at almost every iteration, with at most a 1.2% decline relative to any point, and validation performance on GSM8K drops at most 0.8% from any iteration after improving past few-shot performance (3.1%). Ucno asked specifically about the cause of the difference between the arithmetic curves with and without rationalization. The accuracy for different lengths will improve at the same time to the extent that (some) longer problems are solved and added to the training data. We hypothesize that this happens more often when rationalization is used simply because it is easier to generate a scratchpad from the input numbers and final answer: all tokens in the scratchpad, aside from the carry token, are present in the prompt and the task is mostly to rearrange these tokens. This allows rationalization to generate some longer-length solutions earlier, which otherwise would not have been present. However, we feel that this could be a valuable question to explore more deeply in later work.

Related to the question of training on incorrect rationales, Reviewer Gy9w asks about the frequency of examples which are “right for the wrong reasons” - this is a fundamentally subjective question, so we conducted a human evaluation (discussed in Section 4.4) to investigate the relative preferences of humans of the generated rationales to crowdsourced rationales and few-shot rationales and found that crowdworkers tended to prefer rationales generated by the finetuned model.

As suggested by Reviewer UQiZ, we have added some discussion in Appendix G (“Further Limitations and Impacts”) about whether our techniques will generalize to larger models: On the one hand, it is impossible to know based on our results. On the other, the recent results from PaLM [39] and Minerva [40] suggest that the benefit of rationales increases with scale, so it is not unreasonable to expect that fine-tuning the models on successful rationales will be beneficial. As also suggested by UQiZ, we have added a discussion of the related prior work on iterated learning and weak supervision in NLP including Zettlemoyer and Collins 2009, Guu et al 2017, and Goldman et al 2018.

Finally, we would like to again thank the reviewers for their time and detailed comments. We hope that the above responses and changes to the paper address any concerns and are happy to clarify any further details.

---

> ### Author Response · Authors · 2022-08-05
> **Experiment follow-up discussion**
>
> Thanks again to the reviewers for their constructive and positive feedback. Following up on the additional experiments, we found that the model's performance on the CQA test set was 64.9%. This is about what one might expect, as the CQA test set is known to be more difficult than the dev set, with LLM in-context performance (e.g. GPT3) unavailable but other CQA models (e.g. T5, RoBERTa, XLNet, ALBERT, DREAM (ensemble)) losing about 5-8%.
>
> As also promised, here are the additional ablation results, to be incorporated into the text:
> * Rationalization-only training on CQA reaches 69.2%, which is higher than rationale generation without rationalization (68.8%), but lower than the combination (72.5%).
> * Continual training on CQA reaches 70.6% accuracy
> * Temperature sampling (at t=0.5) on CQA reaches 72.0% accuracy
>
> The drop from rationalization-only training is not necessarily unexpected, as it is presumably easier for the model to produce the right answer with a bad rationale if it is told the correct answer in advance. The drop from continual training mostly matches our expectations. The slight decrease from temperature sampling is potentially caused by the factors discussed in Section 5, but it is less substantial than in our preliminary results from before we incorporated few-shot training. We will update the discussion of Temperature in Section 5 to reflect these details.
>
> We look forward to answering any further questions!

---

### Meta-Review · Area_Chair_iDuA · 2022-08-31

**Recommendation:** Accept
**Confidence:** Certain

**Metareview:**

All reviewers found this paper to be interesting and timely on a topic of much current interest -- prompting LLMs with rationales to improve their accuracy. While methodologically the paper follows existing work in not so novel ways, the proposed procedure leads to strong enough empirical gains to be of wide interest to the community. The authors are encouraged to revise the paper incorporating the reviewer's suggestions.

**Award:**

No

---

### Decision · Program_Chairs · 2022-09-14

Accept